# Mechanically driven Li dendrite penetration in garnet solid electrolyte

Yuwei Zhang[1✉], Soroush Motahari[1], Eric V. Woods[1], Stefan Zaefferer[1], Peter Schweizer[1], Zhiyuan Zhang[1], Yuqi Liu[1], Baptiste Gault[1,2], Franz Roters[1], Dierk Raabe[1], Christina Scheu[1], Yug Joshi[1], Siyuan Zhang[1✉], Chuanlai Liu[1,3✉] & Gerhard Dehm[1✉]

All-solid-state batteries promise improved safety and higher energy density by replacing flammable liquid electrolytes and graphite anodes with solid electrolytes and lithium metal[1–4]. However, the penetration of soft lithium dendrites into hard ceramic electrolytes remains a substantial obstacle to realizing all-solid-state lithium metal batteries[5–7]. The mechanism by which mechanically soft lithium dendrites fracture hard ceramic electrolytes remains under debate[7–10] owing to the challenges of characterizing nanoscale lithium distribution and its microstructure at the dendrite tip[11]. Here we investigate the fracture process driven by lithium dendrites in garnet electrolytes using multiscale cryogenic electron microscopy and micromechanical fracture models. We directly visualize lithium dendrites fully filling nanoscale crack tips and extending into micrometre-scale cracks. Limited crystal lattice rotation and plasticity in lithium dendrites indicate that the plated lithium generates substantial hydrostatic stress, which induces tensile stress in the solid electrolyte and drives both intergranular and transgranular fracture. By contrast, the region ahead of the lithium dendrite tip shows no measurable enrichment of lithium or lithium metal nuclei. The mechanically driven lithium penetration in garnet solid electrolyte can be redirected by geometrically engineered voids in the electrolyte, thus mitigating short-circuiting. Our findings suggest that grain boundary toughening and defect engineering are effective strategies for designing dendrite-resistant solid electrolytes.

The counterintuitive phenomenon that soft lithium penetrates hard ceramic solid electrolytes has been attributed to two distinct failure mechanisms. One proposes that internal pressure build-up within lithium dendrites induces mechanical fracture of the solid electrolyte, enabling dendrite propagation and eventual short-circuiting[6–8,12–18]. The other suggests that electron leakage along grain boundaries of solid electrolytes promotes the formation of isolated lithium nuclei, which subsequently interconnect and short-circuits the cell[9,10,19–21]. Resolving the mechanism of the 'soft-penetrates-hard' phenomenon requires microstructural and chemical information of lithium at both the nanoscale and the microscale, especially at the dendrite tip, where lithium deposition and crack propagation occur.

Using a model cell design and a suite of instruments for cryogenic electron microscopy, here we report both intergranular and transgranular fracture events in the solid electrolyte, with lithium fully filling nanoscale cracks at the dendrite tip. No isolated lithium nuclei were detected ahead of the dendrite tip by cryogenic scanning transmission electron microscopy (cryo-STEM). Notably, small crystal lattice rotations were observed in the regions of lithium dendrite adjacent to the $Li_{6.6}La_3Zr_{1.6}Ta_{0.4}O_{12}$ (LLZTO) interface, whereas the dendrite interior exhibited no measurable lattice rotation, indicating a nearly shear-free and thus largely hydrostatic stress state within the lithium dendrite. This interpretation was further supported by micromechanical fracture modelling. On the basis of the mechanically driven lithium dendrite penetration in garnet solid electrolyte, we propose a mechanics-informed strategy to redirect lithium dendrite propagation through the introduction of geometrically engineered voids in LLZTO.

## Intergranular and transgranular dendrite growth

To directly characterize the penetration of lithium dendrite in the solid electrolyte, we used an in-plane cell geometry with a mechanically thinned (about 150 μm) LLZTO pellet (Fig. 1a). This configuration promotes the formation of a single, macroscopically straight, through-thickness dendrite initiated from a tungsten needle, under an overpotential of approximately 50 mV versus Li/Li+ during electrochemical biasing (Fig. 1b). The in-plane thin pellet allows unambiguous identification of the dendrite tip under an optical microscope, in contrast to a thick pellet (Supplementary Fig. 1), and facilitates site-specific cryogenic characterization at the dendrite tip.

Supplementary Fig. 2 shows the infrastructure, which includes a cryogenic scanning electron microscope and focused ion beam (cryo-FIB)

[1]Max Planck Institute for Sustainable Materials, Düsseldorf, Germany. [2]Groupe de Physique des Matériaux, UMR 6634, Université de Rouen Normandie, CNRS, INSA Rouen Normandie, Rouen, France. [3]National Engineering Research Center of Light Alloy Net Forming and State Key Laboratory of Metal Matrix Composite, School of Materials Science and Engineering, Shanghai Jiao Tong University, Shanghai, China. ✉e-mail: yuwei.zhang@mpi-susmat.de; siyuan.zhang@mpi-susmat.de; c.liu@mpi-susmat.de; dehm@mpi-susmat.de

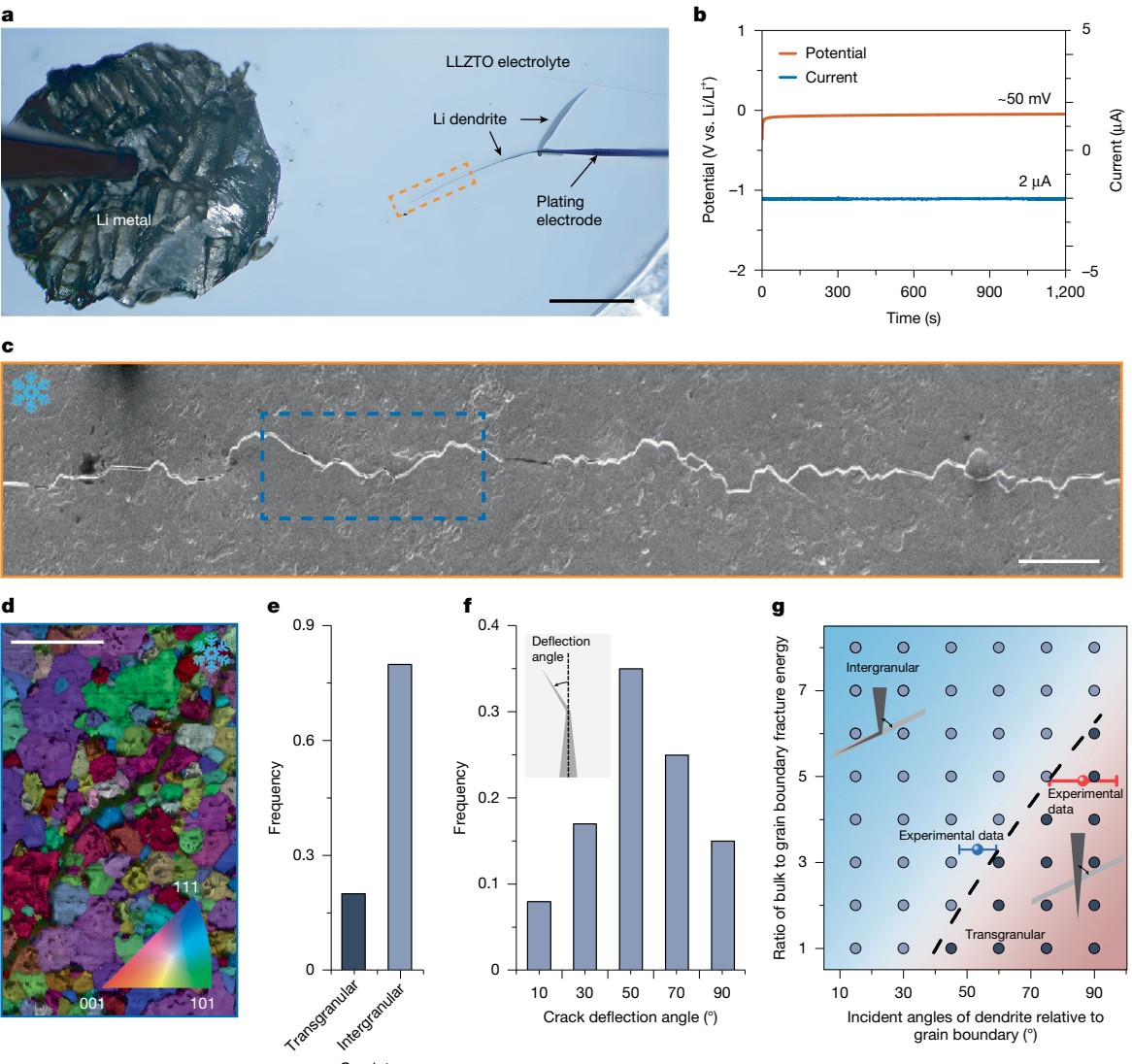

**Fig. 1 | Morphology, microstructure and fracture statistics of LLZTO solid electrolyte during lithium dendrite penetration. a**, Lithium dendrite growth within LLZTO observed using an in-plane cell geometry, which promotes the formation of sharp, straight, through-thickness dendrites. **b**, Corresponding potential and current profiles during lithium plating. **c**, Post-mortem cryo-SEM image showing a tortuous fracture path induced by dendrite growth. **d**, EBSD map of the fractured region of LLZTO under in-plane geometry view, highlighting a mixture of intergranular and transgranular fracture modes. **e**, Fractions of intergranular and transgranular fractures within LLZTO induced by lithium plating. **f**, Histogram of intergranular crack deflection angles, showing frequent high-angle deflections. **g**, Fracture mode map based on phase-field modelling, showing the transition from transgranular to intergranular fracture as a function of deflection angle and the ratio of fracture energy between grain interior and grain boundary. The fitted experimental data with error bars are taken from Fig. 1f and Supplementary Fig. 4b. Scale bars, 1 mm (**a**); 10 μm (**c**,**d**).

system connected to a glovebox by means of an ultrahigh-vacuum transfer suitcase[22]. An inert gas transfer holder enables the transfer of samples at room temperature from the glovebox to the cryogenic scanning transmission electron microscope. Figure 1c shows a cryogenic scanning electron microscopy (cryo-SEM) image of fracture patterns in the LLZTO solid electrolyte caused by lithium dendrite penetration. Compared with the macroscopically straight crack path traced by optical microscopy (Fig. 1a), the fracture pattern is tortuous at the microscale (Fig. 1c). A representative electron backscatter diffraction (EBSD) map (Fig. 1d) indicates that this tortuosity results from a frequently alternating combination of intergranular and intragranular fractures within the solid electrolyte. Further EBSD maps from four other regions are provided in Supplementary Fig. 3. As quantified in Fig. 1e, 20 ± 1% of the roughly 100 analysed grains exhibited transgranular cracking. The high fraction of transgranular cracks indicates that lithium dendrite propagation is unlikely to be governed by isolated

lithium nucleation along grain boundaries. Notably, as shown in Fig. 1f, nearly 75% of all intergranular cracks exhibit deflection angles greater than 40° as measured from EBSD, even though such deflection substantially reduces the maximum tensile stress ahead of the dendrite tip[23]. The crack deflection angle measured directly from Fig. 1c yields a similar result (Supplementary Fig. 4a). This behaviour suggests that local variations in fracture resistance between grain boundaries and grain interiors play a dominant role in crack deflection. Moreover, statistical analysis (Supplementary Fig. 5) of the intergranular crack occurrence as a function of grain boundary misorientation angles reveals that the misorientation distribution in the cracked solid electrolyte closely mirrors that of the pristine pellet, suggesting that grain boundary misorientation plays only a small role in crack deflection. The lithium-plating-induced fracture behaviour of the lithium/LLZTO/lithium symmetrical cell was also examined, as shown in Extended Data Fig. 1 and Supplementary Fig. 6. Both intergranular and transgranular

fracture features were observed, with approximately 20–30% of the fracture events being transgranular. However, it is not possible to accurately quantify the intergranular–transgranular ratio or determine the crack deflection angle because the fracture contour is not well defined in the symmetrical cell geometry, in contrast to the clearly resolved crack paths shown in Fig. 1d.

To rationalize the observed fracture behaviour, we performed phase-field simulations of lithium-plating-induced crack propagation across grain boundaries, systematically varying both the grain boundary fracture energy and the deflection angle[24,25]. The simulations (Fig. 1g) show that crack deflection occurs when the fracture resistance of the grain boundary is less than half that of the grain interior at a deflection angle of 45°. The calculated stress distribution (Supplementary Fig. 7) shows that the maximum tensile stress ahead of the crack tip reduces substantially from 450 MPa to 337 MPa on crack deflection. The predicted fracture mode map in Fig. 1g delineates the transition between transgranular and intergranular fracture as a function of the crack deflection angle and the fracture energy ratio between the grain interior and the grain boundary. For large deflection angles, the model predicts that intergranular cracking occurs only when the fracture energy of the grain boundary is substantially lower than that of the grain interior. By fitting the experimental data (Fig. 1f and Supplementary Fig. 4b) to the fracture mode map (Fig. 1g), we conclude that the average grain boundary fracture energy is a factor of 3–5 lower than that of the bulk (the error bars represent the 95% confidence interval of the mean).

## Nanoscale Li mapping at the crack tip

To unveil the failure mechanism underlying the soft-penetrates-hard phenomenon, we investigated the microstructure and lithium distribution at the dendrite tip with nanoscale resolution using cryo-STEM and cryogenic electron energy loss spectroscopy (cryo-EELS). As shown in Fig. 2a, we prepared STEM lamellae near the dendrite tip from three regions: (1) a plan-view slice at the dendrite tip; (2) a cross-sectional view at the dendrite tip; and (3) a region approximately 1 μm ahead of the dendrite tip. Three-dimensional reconstruction of the crack network near the dendrite tip (Fig. 2b) and pristine LLZTO solid electrolyte (Supplementary Fig. 8) were performed. Owing to the susceptibility of lithium and LLZTO to degradation from ambient air and electron beam exposure, all sample preparation and STEM observations were performed under vacuum at cryogenic temperatures[11].

Cryo-FIB serial sectioning revealed a three-dimensional tortuous crack path through the solid electrolyte (Fig. 2b), reflecting the intertwined intergranular and transgranular fracture modes as observed in the surface analyses[26] (Fig. 1c,d). Further cross-sectional SEM images taken at several locations near the dendrite tip are shown in Supplementary Fig. 9. In the lithium/LLZTO/lithium symmetrical cell configuration, the three-dimensional fractography of LLZTO was also further analysed, as presented in Fig. 2h. The electrochemical data that pertain to the growth of lithium dendrites through the symmetrical cell are shown in Supplementary Fig. 10. A relatively high current density was intentionally applied, under which lithium dendrite nucleation and growth are expected to occur. A sharp increase in overpotential (red curve in Supplementary Fig. 10a) developed after approximately 2,500 s of plating, which is attributed to contact loss on the stripping side[27]. In the subsequent cycle (blue curve in Supplementary Fig. 10b), after current reversal, lithium metal preferentially deposits at pre-existing 'hotspots', leading to accelerated dendrite growth[12]. Once dendrites begin to grow within the solid electrolyte, the effective transport distance for Li+ ions is reduced, which explains the observed decrease in overpotential with increasing time in Supplementary Fig. 10b. At the end of the experiment, we intentionally terminated the test before a hard short circuit, unlike the case shown in Extended Data Fig. 1b. This precaution was taken because a hard short circuit may induce localized Joule heating, which might melt lithium dendrites and alter the microstructure of

the lithium dendrites[28]. For the lithium dendrite growth through the symmetrical cell configuration, the reconstructed lithium dendrite morphology shown in Fig. 2h and Supplementary Fig. 11 exhibits a tortuous geometry similar to that shown in Fig. 2b. Notably, the seemingly isolated lithium dendrites #1, #2 and #3 (Fig. 2i, slice 350) originate from a single, relatively continuous dendrite (Fig. 2i, slice 1). Without three-dimensional reconstruction at cryogenic temperature, such an observation could be misinterpreted as evidence for isolated lithium nucleation and propagation within the LLZTO electrolyte. Directly imaging the lithium dendrite within the fractured LLZTO is challenging owing to the low yield of secondary electrons in this region, which causes plated lithium to appear similar to vacuum in SEM.

To map the lithium distribution at the dendrite tip, we performed cryo-STEM-EELS mapping on the lamellae prepared at the dendrite tip in both plan and cross-sectional views (Fig. 2c,d). Notably, the dendrite tip on the lower right side of Fig. 2c, initially sharp before trench milling, becomes wider after lamella preparation, probably because of the release of mechanical constraint following removal of surrounding material. The STEM contrast in Fig. 2c,d clearly indicates that the crack tip is fully filled. EELS mapping in Fig. 2f,g further confirms that lithium fully fills the nanoscale dendrite tip. The Li K-edge fine structures are differentiated using multivariate statistical analysis and shown in Supplementary Fig. 12 (ref. 29). For comparison, in the region 1 μm ahead of the main dendrite, Fig. 2e shows only dark diffraction contrast at grain boundaries (unlike the bright contrast from lithium observed in Fig. 2c,d), despite this region being commonly considered one of the most energetically favourable sites for lithium nucleation[9,16,30,31]. The chemical homogeneity of this region is further confirmed by energy-dispersive X-ray spectroscopy performed in STEM mode (Supplementary Fig. 13). Previous work by Liu et al.[9] reported that the bandgap at the grain boundaries of garnet solid electrolytes seems to be lower than that of the bulk, enabling lithium nucleation at triple junctions during in situ TEM biasing at 10 V versus Li/Li+, whereas no lithium nuclei were observed at lower biases of 2 V and 5 V. In Supplementary Fig. 14, we show an extra STEM lamella prepared ahead of the dendrite tip. The dendrite was grown under an overpotential of 50 mV versus Li/Li+. No EELS signal corresponding to lithium was detected within the triple junctions (Supplementary Fig. 14). These results not only agree with the observations of Liu et al.[9] at lower bias but also suggest that, within the normal battery operating voltage window[4] (<4.5 V versus Li/Li+), lithium plating does not produce measurable isolated lithium accumulation at grain boundaries or triple junctions in LLZTO electrolytes. Beyond the filling condition inside the crack, no dislocation activity was observed near the fractured LLZTO, indicating the absence of stress relaxation through plasticity, confirming the brittle nature of the garnet-type electrolyte[32,33]. This finding emphasizes the need to develop mechanically tougher ceramic electrolytes to accommodate stress accumulation during lithium dendrite penetration[34–36].

Overall, cryo-SEM and cryo-STEM observations show that lithium fully occupies the nanoscale crack at the dendrite tip. By contrast, lithium enrichment was not observed ahead of the dendrite tip within nanoscale resolution, including grain boundaries and triple junctions, which could otherwise serve as nuclei for dendrite propagation. Despite our evidence supporting a mechanically governed mechanism for dendrite propagation in garnet solid electrolyte, an important question remains: how can soft lithium generate sufficient internal stress to fracture a stiff ceramic electrolyte? To address this, we next examine the stress state and plastic activity of lithium dendrites confined within cracks, using grain orientation mapping through EBSD and micromechanical fracture modelling.

## What drives soft-penetrates-hard?

To investigate the driving force behind lithium dendrite penetration in solid electrolytes, we analysed the microstructure and stress state of

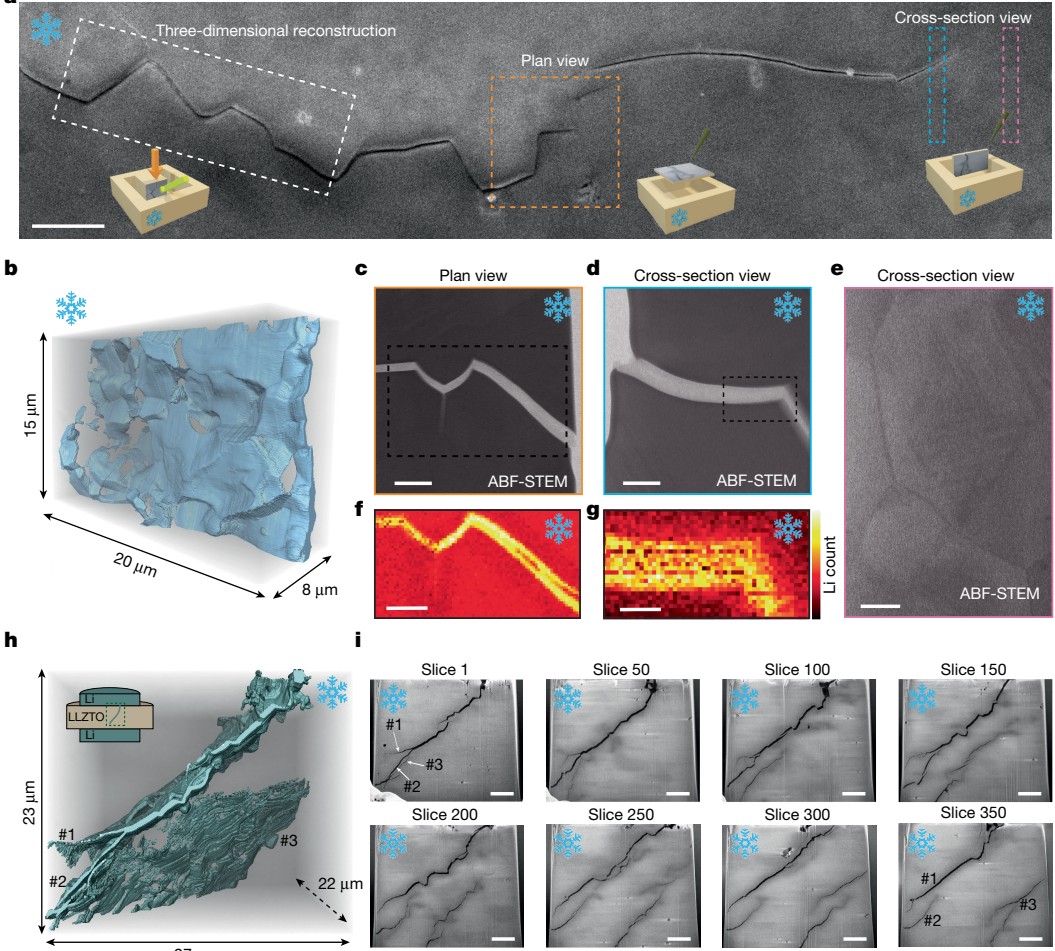

**Fig. 2 | Fractography and elemental distribution at the lithium dendrite tip.**
**a**, Cryo-SEM image showing the location of the lithium dendrite tip in the
LLZTO solid electrolyte and the regions selected for STEM lamella preparation,
including areas for three-dimensional reconstruction and plan-view and cross-
section observations. The dendrite seen here was grown at 2 μA. **b**, Three-
dimensional reconstruction of the crack network near the dendrite tip
obtained by cryo-FIB serial sectioning. **c**,**d**, Plan-view (**c**) and cross-section (**d**)
observations of the lithium dendrite within the LLZTO electrolyte,
corresponding to the regions marked in **a**. **e**, Cross-section observation
of a region located approximately 1 μm ahead of the dendrite tip.
**f**,**g**, Corresponding EELS lithium count maps for the boxed regions in **c**,**d**,

confirming lithium presence at the crack tip. **h**, Three-dimensional
reconstruction of lithium dendrites in the solid electrolyte grown in the
symmetric cell configuration, as illustrated by the schematic. **i**, Selected
cross-section slices used for the three-dimensional reconstruction shown
in **h**. Labels #1, #2 and #3 mark three seemingly isolated dendrites in slice
350, which originally branched from a single main dendrite, as shown in
slice 1 and panel **h**. Note that Fig. 2h,i were acquired from the symmetric cell
configuration, in contrast to Fig. 2a–g. ABF-STEM, annular bright-field
scanning transmission electron microscopy. Scale bars, 2 μm (**a**); 500 nm
(**c**,**e**,**f**); 250 nm (**d**); 100 nm (**g**); 5 μm (**i**).

lithium dendrites confined within LLZTO. We selected a region in which
the dendrite widens to the micrometre scale, enabling characterization
by means of cryo-STEM and cryogenic transmission Kikuchi diffraction
SEM (cryo-TKD-SEM). Unlike the single sharp crack tip observed in
Fig. 2, this region, located away from the main dendrite tip, exhibits a
dense network of nanocracks emanating from the main lithium dendrite
(Fig. 3a). These branched nanocracks resemble the localized dendrite
tip shown in Fig. 2c,d. Cryo-EELS mapping confirms that these nanoc-
racks are also fully filled with lithium metal (Supplementary Fig. 15).

Figure 3b shows the lattice orientation map of the lithium dendrite
within the crack, characterized by cryo-TKD-SEM, with further results in
Supplementary Fig. 16. The lithium dendrite exhibits an average grain
size of 5 μm, a value substantially smaller than that of bulk lithium metal,
in which the grain size typically exceeds hundreds of micrometres[37–39].
Furthermore, the grain reference orientation deviation (GROD) angles
map in Fig. 3c,d reveals minimal lattice rotation across most of the den-
drite volume. Moderate orientation gradients are observed only near
the lithium–LLZTO interface, whereas the dendrite interior exhibits

only minimal gradients, which may be interpreted as the absence of
plastic strain. The microstructure of the lithium dendrite growth within
the solid electrolyte under the symmetrical cell configuration is shown
in Supplementary Fig. 17. The GROD map also exhibits minimal lattice
rotation, comparable with that observed in Fig. 3c. These observations
together indicate that lithium dendrites undergo negligible plastic
deformation during their penetration through the ceramic electrolyte.

We performed phase-field fracture modelling to simulate lithium
dendrite growth within LLZTO. Figure 3e shows the evolution of the
stress and plastic strain fields as the lithium dendrite penetrates the
solid electrolyte, assuming a lithium yield strength of 125 MPa (ref. 40).
The simulated stress distribution ahead of the propagating lithium den-
drite (Supplementary Fig. 18) closely matches previous experimental
measurements[41]. Lithium plating within the confined crack (Fig. 3f)
leads to a build-up of a high hydrostatic stress state in the lithium,
reaching around 600 MPa. This internal pressure translates into high
tensile stress of comparable magnitude in the surrounding electrolyte,
which in turn drives crack propagation and further lithium intrusion.

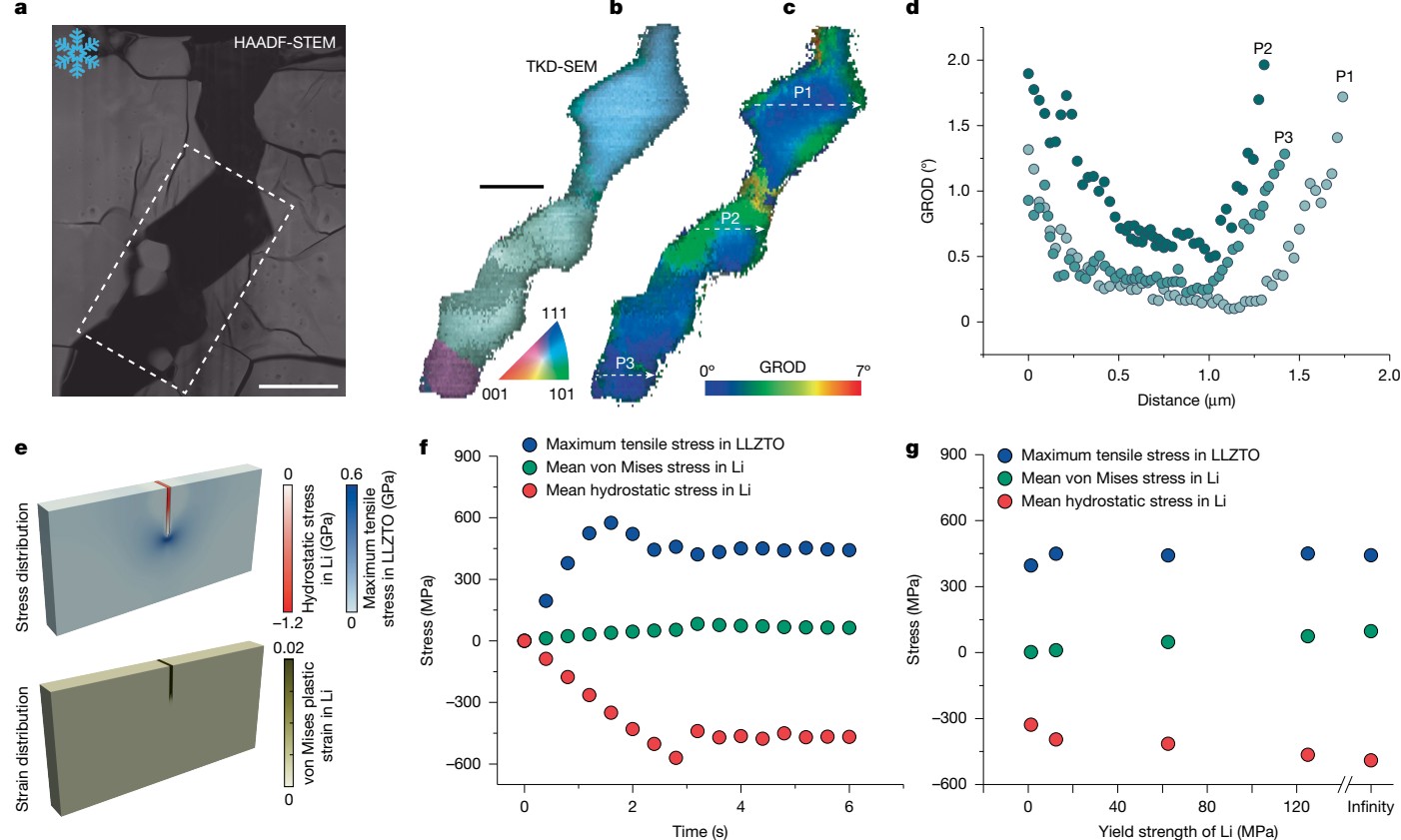

**Fig. 3 | Microstructure of lithium dendrite in LLZTO and phase-field fracturing modelling of lithium dendrite penetration. a**, High-angle annular dark-field scanning transmission electron microscopy (HAADF-STEM) image showing a lithium dendrite within the solid electrolyte, with nanocracks branching from the main dendrite. **b,c**, TKD-SEM maps showing the grain orientation (**b**) and GROD (**c**) within the lithium dendrite. **d**, Point-to-origin misorientation profiles along the three lines marked in **c**, showing negligible crystal lattice rotation across the dendrite except the regions near the lithium–LLZTO interface. **e**, Simulated evolution of the maximum tensile stress in LLZTO, hydrostatic stress in lithium and von Mises plastic strain in lithium in three dimensions. The initial crack length is equal to 24 μm. **f**, Stress evolution during lithium dendrite propagation. **g**, Stress response in lithium and LLZTO as a function of lithium yield strength ranging from 1 MPa to 125 MPa and a purely elastic case, confirming hydrostatic stress as the dominant fracture-driving force. von Mises plastic strain: a scalar measure of the accumulated plastic deformation derived from the plastic strain tensor. GROD, grain reference orientation deviation. Scale bars, 2 μm (**a,b**).

The plastic strain develops accordingly, following the crack propagation; however, notably, the von Mises stress remains one order of magnitude below the hydrostatic stress, once again indicating that plastic deformation in lithium dendrite is limited. Extended simulation results are shown in Supplementary Figs. 19–24.

Given that the yield strength of lithium is strongly dependent on size[34,39,40,42,43], we performed simulations ranging from 1 MPa to 125 MPa, including the case of a purely elastically deforming lithium, as shown in Fig. 3g. The results show that the hydrostatic stress within the lithium dendrite remains orders of magnitude higher than the von Mises stress, regardless of yield strength. Plastic deformation is localized near the lithium–LLZTO interface, whereas most of the dendrite remains largely plastically undeformed. These findings again confirm that the hydrostatic stress, rather than plasticity, acts as the dominant driving force behind the soft-penetrates-hard phenomenon during lithium dendrite growth.

## Redirecting Li dendrite growth

On the basis of the mechanically driven mechanism of lithium dendrite penetration in garnet solid electrolyte, we propose a mitigation strategy that uses cracks to redirect lithium dendrite growth in a transverse direction (Fig. 4a), aiming to prevent the formation of a short circuit. Two arrays of Vickers indents (Fig. 4a,c) were introduced perpendicular to the anticipated dendrite propagation path. As shown in Fig. 4a, on application of an electrical bias, lithium began plating within the solid electrolyte. When the lithium dendrite reached the indent arrays, it abruptly deflected macroscopically and adopted a new route (Fig. 4b). The post-mortem cryo-SEM image (Fig. 4d) shows that the lithium dendrite propagates towards the cracks emanating from the Vickers indent and subsequently deflects by approximately 45°, following the pre-existing cracks induced by Vickers indent. The subsurface interaction between the lithium dendrite and the LLZTO solid electrolyte is shown in Supplementary Fig. 25. With continued plating, the dendrite maintained this redirected trajectory without reverting to the original direction (Fig. 4b). By contrast, another lithium dendrite that did not encounter the engineered obstacle continued along its initial direction and ultimately caused a short circuit. Supporting video and an extra example are provided in Supplementary Video 1 and Supplementary Fig. 26. We note that plastic strain accumulation adjacent to the indent is negligible, as EBSD mapping around the indent (both at and below the surface as shown in Supplementary Fig. 27) reveals no measurable orientation gradients and no systematic increase in local misorientation (for example, kernel average misorientation). Moreover, the residual stress in the polycrystalline LLZTO is largely relaxed by cracking and was also measured to be on the order of 1 MPa (Supplementary Figs. 28 and 29). Previous studies[14] have shown that dendrite deflection in garnet electrolyte requires stresses on the order

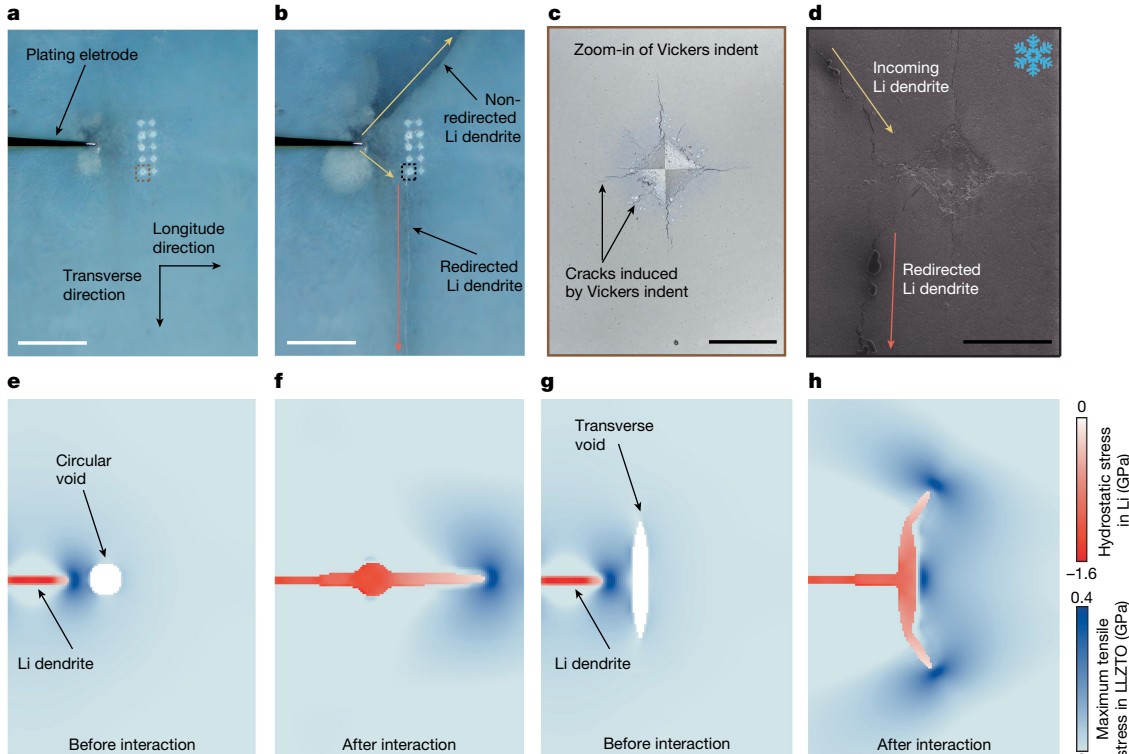

**Fig. 4 | Tailor lithium dendrite growth through engineered voids.**
**a**,**b**, Top-view optical microscope image showing lithium dendrite growth before (**a**) and after (**b**) encountering arrays of Vickers indents. The dendrite deflects on reaching cracks emitting from the Vickers indent, altering its trajectory. **c**, Zoom-in optical microscope image of a representative Vickers indent. **d**, Zoom-in SEM image showing the interaction between the crack network and the lithium dendrite. **e**–**h**, Phase-field simulation of lithium dendrite interaction with circular (**e**) and transverse (**g**) voids embedded in the solid electrolyte, showing hydrostatic stress in lithium and maximum tensile stress in LLZTO during dendrite propagation. Transverse voids promote deflection, whereas circular voids allow straight propagation. Scale bars, 1 mm (**a**,**b**); 50 μm (**c**,**d**).

of 100 MPa, whereas the measured residual stress surrounding our Vickers indents is two orders of magnitude lower. Accordingly, our fracture modelling indicates that this level of stress has a negligible influence on the observed crack deflection (Supplementary Fig. 29d,e). Detailed measurements and calculations are provided in Supplementary Figs. 27–31.

To understand how cracks tailor lithium dendrite growth, we performed phase-field simulations of dendrite–void interactions in LLZTO electrolytes under electrochemical plating conditions. In the model, transverse voids were used to represent the cracks introduced by the Vickers indents. Both circular and transverse voids were modelled to reflect the mechanical environment experienced by a growing lithium dendrite. Consistent with our experimental observations, the simulations show that lithium dendrites are deflected on encountering transverse voids (Fig. 4g,h). By contrast, dendrites penetrate directly through circular voids without deflection (Fig. 4e,f), demonstrating that void geometry plays a critical role in guiding dendrite paths. For transverse voids, lithium confinement within voids generates localized hydrostatic pressure, which induces asymmetric tensile stress concentrations at the electrolyte–void interface. These stresses promote crack deflection along paths tangential to the major axis of the void. By contrast, when lithium fills a circular void, it results in tensile stress concentration aligned with the original propagation path, thereby favouring continued straight penetration of the dendrite. These results show that transverse voids could redirect the growth direction of lithium dendrites by locally modifying the tensile stress distribution within the ceramic electrolyte. This mechanistic understanding offers a proof of concept for implementing defect-based strategies to control dendrite propagation and suppress short-circuiting in solid-state batteries.

## Implications to battery design

This study shows that lithium dendrite penetration into garnet solid electrolytes is driven by mechanically induced fracture, as schematically illustrated in Supplementary Fig. 32. To suppress dendrite-induced failure and enable reliable solid-state lithium metal batteries, we propose the following design strategies based on our mechanistic insights:

1. Enhancing grain boundary fracture resistance: as shown in Fig. 1g, cracks often deflect along grain boundaries, even at the expense of a reduced driving force for crack propagation. This behaviour reflects insufficient fracture resistance along grain boundaries (a factor 3–5 times weaker than the fracture resistance of bulk). Strategies such as doping have been reported to strengthen grain boundaries[44,45].

2. Improving the fracture toughness of the solid electrolyte: as shown in Fig. 2c,d, the absence of dislocation activity near the dendrite tips highlights the intrinsically brittle nature of garnet electrolytes and their limited ability to relax stress through plastic deformation during lithium dendrite penetration. Enhancing fracture toughness through mechanisms such as dislocation activation or shear flow can promote stress dissipation and delay crack propagation during lithium plating[35,46,47].

3. Mechanically guided redirection of dendrite propagation: as shown in Fig. 4b, transverse voids aligned perpendicular to the dendrite propagation direction redirect dendrite growth paths and thus prevent short-circuiting. This proof of concept shows that introducing local defects (such as voids, cracks or weak interfaces) can effectively influence the propagation paths of the dendrites. To realize this concept in thin solid electrolyte separators (ideally down to about

20 μm), interfaces in multilayer solid electrolytes could potentially be used as mechanically weak regions to redirect dendrite propagation[48–50]. This approach provides a fundamental design principle for mitigating dendrite-induced short-circuiting while preserving overall ionic transport and chemical stability. Scalable fabrication in such lower-dimensional engineering systems requires further investigations.

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

# Methods

## Li dendrite growth in solid electrolytes

LLZTO solid electrolytes were obtained from Toshima Manufacturing Co., Ltd. The pellets were mechanically ground to a thickness of approximately 150 μm, with final polishing performed using a 0.05-μm alcohol-based colloidal silica suspension. To reduce the interfacial resistance between lithium and the solid electrolyte, the thin LLZTO discs were immersed in 1 M HCl for 30 s to remove surface contaminants, following the procedure demonstrated in ref. 51. Immediately after the acid treatment, the solid electrolyte discs were transferred into an argon-filled glovebox ($O_2$ and $H_2O$ < 0.5 ppm). Lithium foil (MaTeck Material Technologie & Kristalle GmbH) was scraped using a plastic tweezer to expose a fresh, shiny surface. A 3-mm-diameter lithium pad was then punched out and stuck to the LLZTO disc. The assembled cell was placed on a hotplate and baked at 130 °C for 1 h.

To study the interaction between lithium dendrites and specific features of interest, a single lithium metal pad was used as the counter electrode. A tungsten probe was placed on the surface of the solid electrolyte to serve as the working electrode, where lithium dendrites nucleated and grew[41]. A constant current was applied between the lithium metal pad and the tungsten needle using a SP-200 potentiostat (Bio-Logic Science Instruments GmbH). Electrochemical impedance spectroscopy (EIS) data were recorded in the frequency range between 10 Hz and 7 MHz with an amplitude of 50 mV using a SP-200 impedance analyser (Bio-Logic). The growth of lithium dendrites was conducted entirely within the glovebox and monitored using a camera mounted on a stereo microscope (KERN & SOHN GmbH).

Lithium dendrite growth through the symmetric cell configuration was cycled using the same potentiostat equipped with a pressure stand (Imada Inc.). Before applying the bias, the symmetric cell was heated to 130 °C using a heating sleeve (RS Components Ltd.) to improve the interfacial contact between the lithium metal and the solid electrolyte. After short-circuiting, the lithium metal was removed using sandpaper with a grit size of 1,200. The short-circuited solid electrolyte was then taken out of the glovebox, soaked in epoxy overnight for curing and subsequently polished to the region in which features resembling lithium dendrites could be observed, as shown in Extended Data Fig. 1f,g. Extended Data Fig. 1e schematically illustrates the sample preparation procedure. Extended Data Fig. 1f shows the surface after rough polishing with 320-grit sandpaper and Extended Data Fig. 1g shows the result after fine polishing using a 0.1-μm $SiO_2$ polishing suspension. The lithium metal on the plating side, where dendrite growth occurred, could be easily peeled off by hand, as shown in Supplementary Figs. 10, 11 and 17. Therefore, no sandpaper was used to remove the lithium electrode, in contrast to the procedure used for the samples shown in Extended Data Fig. 1 and Supplementary Fig. 6.

## Cryogenic FIB, SEM and EBSD

Using an inert high-vacuum (< $10^{-7}$ mbar) cryogenic transfer suitcase (Ferrovac AG), hereafter referred to as the 'suitcase', the LLZTO disc was transferred from the argon-filled glovebox to a Thermo Fisher Scientific Helios 5 CX Ga FIB/SEM system. The Helios 5 is equipped with an Aquilos cryo-stage featuring free rotation capability and a Thermo Fisher Scientific EZ-Lift tungsten cryogenic micromanipulator. Both the cryo-stage and the manipulator were maintained at −190 °C using active heating control and a nitrogen flow rate of 190 mg s$^{-1}$. All operations inside the FIB/SEM system—including SEM imaging, FIB cutting, TEM lamella preparation and EBSD—were conducted at a stable temperature of −190 °C. The TEM lamella was welded onto both the micromanipulator needle and a copper grid by means of redeposition induced by line cuts, as shown in Supplementary Fig. 33. Detailed lamella preparation procedures have been described in previous works[52,53]. Once thinned to below 150 nm, the lamella and the bulk sample were transferred back into the argon glovebox using the suitcase. The interaction between

the electron beam and the solid electrolyte is strongly suppressed at cryogenic temperatures. No electron-beam-induced lithium nucleation was observed under cryogenic conditions, in contrast to the artefacts frequently encountered at room temperature[11,54].

EBSD patterns of the LLZTO pellet were collected at cryogenic temperature (−190 °C) using a direct electron detector (Clarity Plus, EDAX LLC). Kikuchi patterns were acquired under an accelerating voltage of 10 kV and a beam current of 2.8 nA. To analyse diffraction from a lithium dendrite within the solid electrolyte, a lamella was prepared following the same procedure described above, except the final lamella thickness was maintained at approximately 1 μm. Supplementary Fig. 34 shows the TKD lamella, which maintains its mechanical integrity without any observable bending or distortion induced by ion-milling preparation. Moreover, because the sample was prepared using Ga$^+$ FIB at cryogenic temperature, strain rearrangement during ion milling is expected to be strongly suppressed and therefore experimentally negligible, as reported in several previous studies[37,38,55]. TKD patterns of the lithium dendrite were also acquired using the same direct electron EBSD detector. The diffraction patterns of the lithium dendrite were analysed using spherical indexing[56]—a new technique that enables improved pattern recognition and orientation determination for low-symmetry or low-quality patterns. In contrast to the classical analysis technique that uses a Hough transform for detection of the Kikuchi bands in Kikuchi patterns[57], spherical indexing is an advanced image matching technique, in which the experimental pattern is compared with a theoretical master pattern. The comparison is done by developing both the experimental and the master pattern into a series of spherical harmonic functions and comparing them by a spherical cross-correlation function. Because spherical indexing matches the whole pattern, this technique can be applied very robustly with weak diffraction patterns, typically obtained from lithium. Furthermore, because the master pattern can be calculated for any diffraction voltage and because the matching is executed directly on the diffraction sphere, the technique is independent of the acceleration voltage of pattern generation and can also be applied to low-voltage patterns. The classical Hough transform, which detects straight lines, fails in this case because of the high curvature of low-energy Kikuchi lines. Spherical indexing, together with the necessary image preprocessing (static and dynamic background subtraction and contrast enhancement) were done using an early build of the software OIM Analysis 9.1 produced by Ametek EDAX. The master pattern was calculated for 10 kV and 20° of sample tilt in transmission. The bandwidth, a parameter that describes the amount of details that is matched in the pattern, was set to 127.

The incident angles between the dendrite and grain boundaries were measured from EBSD results for both intergranular and transgranular fractures. In both cases, the incident angle values follow a normal distribution. The mean values, along with the 95% confidence intervals extracted from Fig. 1f and Supplementary Fig. 4b, were fitted and plotted in Fig. 1g. The positions of the red and blue dots were placed such that their error bars just begin to intersect the boundary between intergranular and transgranular regions, as indicated by the dashed line.

## Cryo-STEM

The STEM lamella was loaded in a Mel-Build holder inside an argon-filled glovebox and then kept under inert argon atmosphere during sample transfer. All analysis was performed at cryogenic conditions (−150 °C). STEM was performed on a Titan Themis microscope (Thermo Fisher Scientific) operated at 300 kV. The aberration-corrected probe has a convergence semiangle of 23.8 mrad. High-angle annular dark-field and annular bright-field STEM micrographs were collected using respective angular ranges of 73–200 and 8–16 mrad. STEM energy-dispersive X-ray spectroscopy spectrum imaging was acquired using a Super-X detector. STEM-EELS spectrum imaging was performed using a Quantum ERS spectrometer (Gatan) with a

collection angle of 35 mrad. To facilitate comparison with EELS spectra reported in the literature, we opt to show raw EELS spectra from selected areas in Supplementary Figs. 12b and 14d,e. Multivariate statistical analysis was performed on the spectrum imaging datasets to separate backgrounds and signals from different lithium-containing phases[29,58,59]. For lithium count maps shown in Fig. 2f,g, power law background was modelled for components 1 and 2, with respective fitting windows of (45, 50) eV and (45, 57) eV. The integration window was kept to (57, 67) eV. As evidenced in Supplementary Fig. 12, the Li K-edge onsets of the LLZTO and the Li/LiOH phases are different. The quantification of lithium is hence facilitated by multivariate statistical analysis[29], for which most of the spatial variance in EELS signal can be expressed in components 1 and 2. As shown in Supplementary Fig. 12c–f, component 1 is mainly located in the dendrite area and the spectral feature is LiOH-like; component 2 relates to the LLZTO area surrounding the dendrite, with LLZTO-like spectral feature. Component 3 no longer resembles a physical spectrum, as it represents small differential signals to modify the two leading components. This observation confirms the dominance of the Li/LiOH and LLZTO phase in this area. Four-dimensional STEM diffraction imaging was recorded using the pixelated detector Electron Microscope Pixel Array Detector (EMPAD, Thermo Fisher Scientific) and a probe convergence semiangle of 0.65 mrad.

## Small-scale mechanical testing on LLZTO solid electrolyte

LLZTO solid electrolytes with a thickness of 1 mm were mechanically ground and polished, with the final step performed using a 0.05-μm alcohol-based colloidal silica suspension. The samples were immersed in 1 M HCl for 30 s to remove surface contaminants. Immediately after the acid treatment, the nanoindentation experiments were conducted using an iMicro Nanoindenter (KLA Inc.) under ambient environment. A constant indentation strain rate of $\dot{\varepsilon} = 0.1\ \mathrm{s}^{-1}$ was applied using a Berkovich diamond pyramidal indenter. Four independent sets of experiments were performed to verify repeatability (Supplementary Fig. 30), with a total testing duration of approximately 30 min. Although a surface carbonate layer can form on LLZTO on exposure to ambient conditions, its thickness within about 0.5 h of air exposure is expected to be negligible compared with the indentation depth[60].

## Phase-field fracture modelling

**Finite strain kinematics.** A microstructural domain $\mathcal{B}_0 \subset \mathbb{R}^3$ undergoing deformation is described by a mapping $\mathcal{X}(\mathbf{x}) : \mathcal{B}_0 \to \mathcal{B}$, which correlates each material point $\mathbf{x} \in \mathcal{B}_0$ to its corresponding position $\mathcal{X}$ within the deformed domain $\mathcal{B}$. The deformation gradient is denoted by $\mathbf{F} = \frac{\partial \mathcal{X}}{\partial \mathbf{x}} = \nabla \mathcal{X}$.

In the present work, the total deformation gradient is multiplicatively decomposed as:

$$\mathbf{F} = \mathbf{F}_e \mathbf{F}_i \mathbf{F}_p, \tag{1}$$

in which $\mathbf{F}_e$ represents the elastic deformation, $\mathbf{F}_i$ captures the deformation induced by the electromechanical reaction (that is, lithium plating) and $\mathbf{F}_p$ accounts for the plastic deformation within the lithium dendrite. The volumetric change from lithium plating, $J_i$, can be given by the following equation:

$$J_i = \det \mathbf{F}_i. \tag{2}$$

We define the plastic and volumetric velocity gradient tensors in the intermediate configurations as $\mathbf{L}_p$ and $\mathbf{L}_i$, respectively. The evolution equations for $\mathbf{F}_p$ and $\mathbf{F}_i$ can be derived as:

$$\dot{\mathbf{F}}_p = \mathbf{L}_p \mathbf{F}_p, \tag{3a}$$

$$\dot{\mathbf{F}}_i = \mathbf{L}_i \mathbf{F}_i. \tag{3b}$$

**Lithium deposition electromechanics.** The volumetric expansion of the solid electrolyte lattice, resulting from the reduction of lithium ions at the reaction site, is captured through the swelling model proposed by Liu et al.[61] and Narayan et al.[62]. The volumetric change is described by the following equation:

$$J_i = 1 + \Omega \eta_{max} \bar{\eta}. \tag{4}$$

Here $\bar{\eta}$ is the normalized quantity of deposited metallic lithium and $\eta_{max}$ denotes the concentration of metallic lithium under fully lithiated conditions. The parameter $\bar{\eta}$ serves as an order parameter that captures the emergence of deposited lithium in the solid electrolyte. The parameter $\Omega$ in equation (4) is the molar volume of lithium and is taken to be constant.

The local volume change rate arising from the deposition of lithium is given by:

$$\dot{J}_i = J_i \mathrm{tr} \mathbf{L}_i, \tag{5}$$

in which tr refers to the trace operation.

Substituting equation (4) into equation (5) results in:

$$\mathrm{tr} \mathbf{L}_i = \frac{\Omega \eta_{max} \dot{\bar{\eta}}}{1 + \Omega \eta_{max} \bar{\eta}}. \tag{6}$$

Provided that the volume expansion occurs isotopically, we can derive the velocity gradient as:

$$\mathbf{L}_i = \frac{h(\bar{\eta})}{3} \frac{\Omega \eta_{max} \dot{\bar{\eta}}}{1 + \Omega \eta_{max} \bar{\eta}} \mathbf{I}, \tag{7}$$

in which $\mathbf{I}$ is a second-order identity tensor. The interpolation function $h$ acts as a regulator to ensure that deposition only takes place in regions in which electrons are available for the reduction process, that is, in the vicinity of the dendrite. The interpolation function $h$ is taken as:

$$h(\bar{\eta}) = \bar{\eta}^3 (6\bar{\eta}^2 - 15\bar{\eta} + 10). \tag{8}$$

**Phase-field fracture model.** In phase-field damage models, sharp cracks are treated as diffuse regions with gradually degraded material properties. This approach eliminates the necessity of explicitly tracking the crack interface. In this work, the energy formulation is based on the Griffith criterion. Therefore, the total free energy can be obtained as follows:

$$\psi = \int_{\mathcal{B}_0} \left( (1 - \varpi) \psi_{LLZTO}^E + \varpi \psi_{Li}^E + \psi_D \right) \mathrm{d}\mathbf{X}, \tag{9}$$

in which $\psi_{LLZTO}^E$ and $\psi_{Li}^E$ delineate the elastic energy density contributions from LLZTO and lithium, whereas $\Psi_D$ accounts for the surface energy density associated with the newly formed crack surfaces. In the phase-field damage model, the order parameter $d \in [0, 1]$ represents the degree of material degradation, in which $d = 1$ corresponds to a fully intact state and $d = 0$ indicates complete material failure. The interpolation parameter $\varpi$ is introduced to distinguish between the energy contributions in the solid electrolyte and the lithiated region of the dendrite within the solid electrolyte.

To ensure that the crack can only initiate and propagate under tensile stresses, the following decomposition is used:

$$\psi_{LLZTO}^E = g(d) \psi_{LLZTO}^{E+} + \psi_{LLZTO}^{E-}. \tag{10}$$

According to the above expression, only the tensile contribution to the energy $\psi_{LLZTO}^{E+}$ is diminished by the degradation function $g(d)$, typically defined as $g(d) = d^2$, whereas the compressive component $\psi_{LLZTO}^{E-}$ remains unaffected.

The equations pertaining to the tensile and compressive components of LLZTO elastic energy can be established through the following relations:

$$\psi_{\text{LLZTO}}^{\text{E}+} = \frac{1}{2}\mathbf{S}_{\text{LLZTO}}^{+} : \mathbf{E}, \tag{11a}$$

$$\psi_{\text{LLZTO}}^{\text{E}-} = \frac{1}{2}\mathbf{S}_{\text{LLZTO}}^{-} : \mathbf{E}, \tag{11b}$$

in which $\mathbf{E} = (\mathbf{F}_e^{\text{T}}\mathbf{F}_e - \mathbf{I})/2$ is the Green–Lagrange strain and $\mathbf{S}_{\text{LLZTO}}^{+}$ and $\mathbf{S}_{\text{LLZTO}}^{-}$ are obtained through the following:

$$\mathbf{S}_{\text{LLZTO}}^{+} = \mathbb{P}^{+} : \mathbf{S}_{\text{LLZTO}}^{0} \tag{12a}$$

$$\mathbf{S}_{\text{LLZTO}}^{-} = \mathbb{P}^{-} : \mathbf{S}_{\text{LLZTO}}^{0}. \tag{12b}$$

Here $\mathbf{S}_{\text{LLZTO}}^{0}$ is the second Piola–Kirchhoff stress within the solid electrolyte. The fourth-order projection tensors $\mathbb{P}^{+}$ and $\mathbb{P}^{-}$, derived within a thermodynamically consistent framework, are formulated as:

$$\mathbb{P}^{\pm} = \frac{\partial \mathbf{S}_{\text{LLZTO}}^{\pm}}{\partial \mathbf{S}_{\text{LLZTO}}^{0}} = \sum_{i=1}^{3}\sum_{j=1}^{3} \frac{\partial \gamma_i^{\pm}}{\partial \lambda_j}\mathbf{n}_i \otimes \mathbf{n}_i \otimes \mathbf{n}_j \otimes \mathbf{n}_j$$
$$+ \sum_{i=1}^{3}\sum_{j\neq i}^{3} \frac{\gamma_i^{\pm} - \gamma_j^{\pm}}{\lambda_i - \lambda_j}\mathbf{n}_i \otimes \mathbf{n}_j(\mathbf{n}_i \otimes \mathbf{n}_j + \mathbf{n}_j \otimes \mathbf{n}_i), \tag{13}$$

in which, for $i = 1, 2, 3$, $\lambda_i$ and $\gamma_i^{\pm}$ correspond to the eigenvalues of $\mathbf{S}_{\text{LLZTO}}^{0}$ and $\mathbf{S}_{\text{LLZTO}}^{\pm}$, respectively. On the other hand, the tangent modulus is computed using the hybrid scheme proposed by Ambati et al.[63], yielding the following expression:

$$\mathbb{C} = g(d)(1 - \varpi)\mathbb{C}_{\text{LLZTO}}^{0} + \varpi\mathbb{C}_{\text{Li}}^{0}. \tag{14}$$

We define the binary interpolation parameter $\varpi$ based on the following relation:

$$\varpi = \begin{cases} 1, & \text{if } g(d)^2\mathbb{C}_{\text{LLZTO}}^{\text{Voigt}} : \mathbb{C}_{\text{LLZTO}}^{\text{Voigt}} \leq \mathbb{C}_{\text{Li}}^{\text{Voigt}} : \mathbb{C}_{\text{Li}}^{\text{Voigt}} \\ 0, & \text{if } g(d)^2\mathbb{C}_{\text{LLZTO}}^{\text{Voigt}} : \mathbb{C}_{\text{LLZTO}}^{\text{Voigt}} > \mathbb{C}_{\text{Li}}^{\text{Voigt}} : \mathbb{C}_{\text{Li}}^{\text{Voigt}}. \end{cases} \tag{15}$$

The surface energy is given by:

$$\psi_{\text{D}} = \frac{\mathcal{G}_{\text{c}}}{l_0}(1 - d) + \frac{1}{2}\mathcal{G}_{\text{c}}l_0|\nabla d|^2, \tag{16}$$

in which $l_0$ is damage characteristic length and $\mathcal{G}_{\text{c}}$ corresponds to the critical energy release rate.

In this study, we assume that cracks within the solid electrolyte remain fully lithiated. This implies that crack propagation and the formation of metallic lithium coincide. This assumption enables the following relationship: $\bar{\eta} = 1 - d$.

By considering isothermal and adiabatic processes, the evolution of the damage order parameter can be derived through an Allen–Cahn type of relation given below:

$$\dot{d} = -M\left[2d\mathcal{H}_{\text{LLZTO}} - \frac{\mathcal{G}_{\text{c}}}{l_0} - \mathcal{G}_{\text{c}}l_0\text{Div}\nabla d\right], \tag{17}$$

in which the parameter $M$ denotes the damage mobility parameter, which controls the rate of damage promotion in the simulation. The history field function $\mathcal{H}_{\text{LLZTO}}$ is introduced to ensure the irreversibility of the damage, which is expressed as:

$$\mathcal{H}_{\text{LLZTO}}(\mathbf{X}, t) = \max_{t \in [0,T]} \psi_{\text{LLZTO}}^{\text{E}+}(\mathbf{E}(\mathbf{X}, t)), \tag{18}$$

in which $\mathbf{E}(\mathbf{X}, t)$ refers to Green–Lagrange strain.

**Mechanical model of lithium.** In the present model, we assume that lithium can undergo isotropic plastic deformation. On plastic deformation, the isochoric response of the material is connected to the deviatoric stress $\mathbf{M}_{\text{dev}}^{\text{p}} = \mathbf{M}_{\text{p}} - \frac{1}{3}\text{tr}\mathbf{M}_{\text{p}}\mathbf{I}$, in which the Mandel stress $\mathbf{M}_{\text{p}}$ serves as the work-conjugate measure to the plastic velocity gradient $\mathbf{L}_{\text{p}}$ and acts as the driving force that governs its evolution.

On the basis of this plasticity model, the strain rate can be computed by the following relation:

$$\dot{\gamma}^{\text{p}} = \dot{\gamma}^0\left(\sqrt{\frac{3}{2}}\frac{\left\|\mathbf{M}_{\text{p}}^{\text{dev}}\right\|_{\text{F}}}{M\xi}\right)^n, \tag{19}$$

in which the internal variable $\xi$ is akin to the slip resistance in the phenomenological crystal plasticity model. In equation (19), $\dot{\gamma}^0$ denotes the reference strain rate and $M$ is the Taylor factor. Consequently, the associated plastic velocity gradient $\mathbf{L}_{\text{p}}$, which operates within the lattice configuration, is expressed as:

$$\mathbf{L}_{\text{p}} = \frac{\dot{\gamma}^{\text{p}}}{M}\frac{\mathbf{M}_{\text{p}}^{\text{dev}}}{\left\|\mathbf{M}_{\text{p}}^{\text{dev}}\right\|_{\text{F}}}. \tag{20}$$

The value of $\xi$ is set to approach a stationary value $\xi_\infty$ asymptotically from its initial value $\xi_0$ according to the following hardening rule:

$$\dot{\xi} = \dot{\gamma}^{\text{p}}h_0\left|1 - \frac{\xi}{\xi_\infty^*}\right|^a \text{sgn}\left(1 - \frac{\xi}{\xi_\infty^*}\right), \tag{21}$$

in which $h_0$ is the initial hardening and $a$ indicates the stress sensitivity exponent. In equation (21), $\xi_\infty^*$ is the modified saturation hardening value and takes the following form:

$$\xi_\infty^* = \xi_\infty + \frac{(\sinh^{-1}(\dot{\gamma}^{\text{p}}/c_1))^{1/c_2}}{c_3(\dot{\gamma}^{\text{p}}/\dot{\gamma}^0)^{1/n}}. \tag{22}$$

This formulation introduces a dependence of the saturation hardening value on the shear strain rate, enabling controlled adjustment through the parameters $c_i$. Last, the Mandel stress $\mathbf{M}_{\text{p}}$ can be related to the second Piola–Kirchhoff stress $\mathbf{S}$ through the following expression:

$$\mathbf{M}_{\text{p}} = \mathbf{F}_i^{\text{T}}\mathbf{F}_i\mathbf{S}. \tag{23}$$

**Stress equilibrium.** The balance of linear momentum requires satisfying the following relation:

$$\text{Div}\mathbf{P} = \mathbf{0}, \tag{24}$$

in which $\mathbf{P}$ is the first Piola–Kirchhoff stress.

### Simulation set-up

The two-dimensional simulation is conducted under plane strain boundary conditions on a $128 \times 256$ computational grip, with no applied external mechanical deformation. A pre-existing notch is introduced at the start to represent imperfections at the lithium anode–solid electrolyte interface. To study lithium-plating-induced crack propagation across grain boundaries, we use a bicrystal geometry in which the grain boundary is assigned a range of fracture energies and deflection angles to account for variations in grain boundary fracture behaviour. To investigate the interaction between lithium dendrite and engineered voids in LLZTO, circular and transverse voids are introduced ahead of the lithium dendrite within the LLZTO electrolyte. To examine the influence of residual stress on lithium dendrite propagation, an external compression stress of 1 MPa is applied to the model (Supplementary

Fig. 29d,e). In all other simulations, no external mechanical loading is applied. Moreover, three-dimensional simulations are performed with a grid of 128 × 256 × 10, in which the pre-existing notch extends through the thickness of the geometry. These three-dimensional simulations (Fig. 3e) verify the two-dimensional results and confirm that the predicted fracture behaviour remains consistent across both geometries.

## Data availability

All data are available in the main text and the Supplementary Information.

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

**Acknowledgements** Y.Z. thanks the discussion with T. Zhu from Georgia Institute of Technology and C. D. Fincher from Massachusetts Institute of Technology and the coding support from Y. Wang. Max Planck Scholarship (Y.Z.). Helmholtz School for Data Science in Life, Earth and Energy (S.M.). IMPRS graduate school SusMet from Max Planck Society (Y.L.). German Research Foundation Leibniz Prize (E.V.W. and B.G.). EU Horizon Research and Innovation Actions grant 101192848 (Y.J., S.M. and C.L.). National Natural Science Foundation of China grant 52571149 (C.L.). Shanghai Jiao Tong University AI for Engineering Initiative (C.L.).

**Author contributions** Conceptualization: Y.Z., C.L., S. Zhang, G.D. Methodology: Y.Z., S.M., E.V.W., S. Zaefferer, P.S., Z.Z., Y.L., Y.J., S. Zhang, C.L., G.D. Investigation: Y.Z., S.M., E.V.W., S. Zaefferer, P.S., Z.Z., Y.L., Y.J., S. Zhang, C.L. Visualization: Y.Z., S.M., C.L., S. Zhang. Funding acquisition: B.G., F.R., D.R., C.S., Y.J., C.L., G.D. Project administration: G.D. Supervision: G.D. Writing – original draft: Y.Z., C.L., S. Zhang, G.D. Writing – review and editing: all authors.

**Funding** Open access funding provided by Max Planck Society.

**Competing interests** The authors declare no competing interests.

**Additional information**
**Correspondence and requests for materials** should be addressed to Yuwei Zhang, Siyuan Zhang, Chuanlai Liu or Gerhard Dehm.

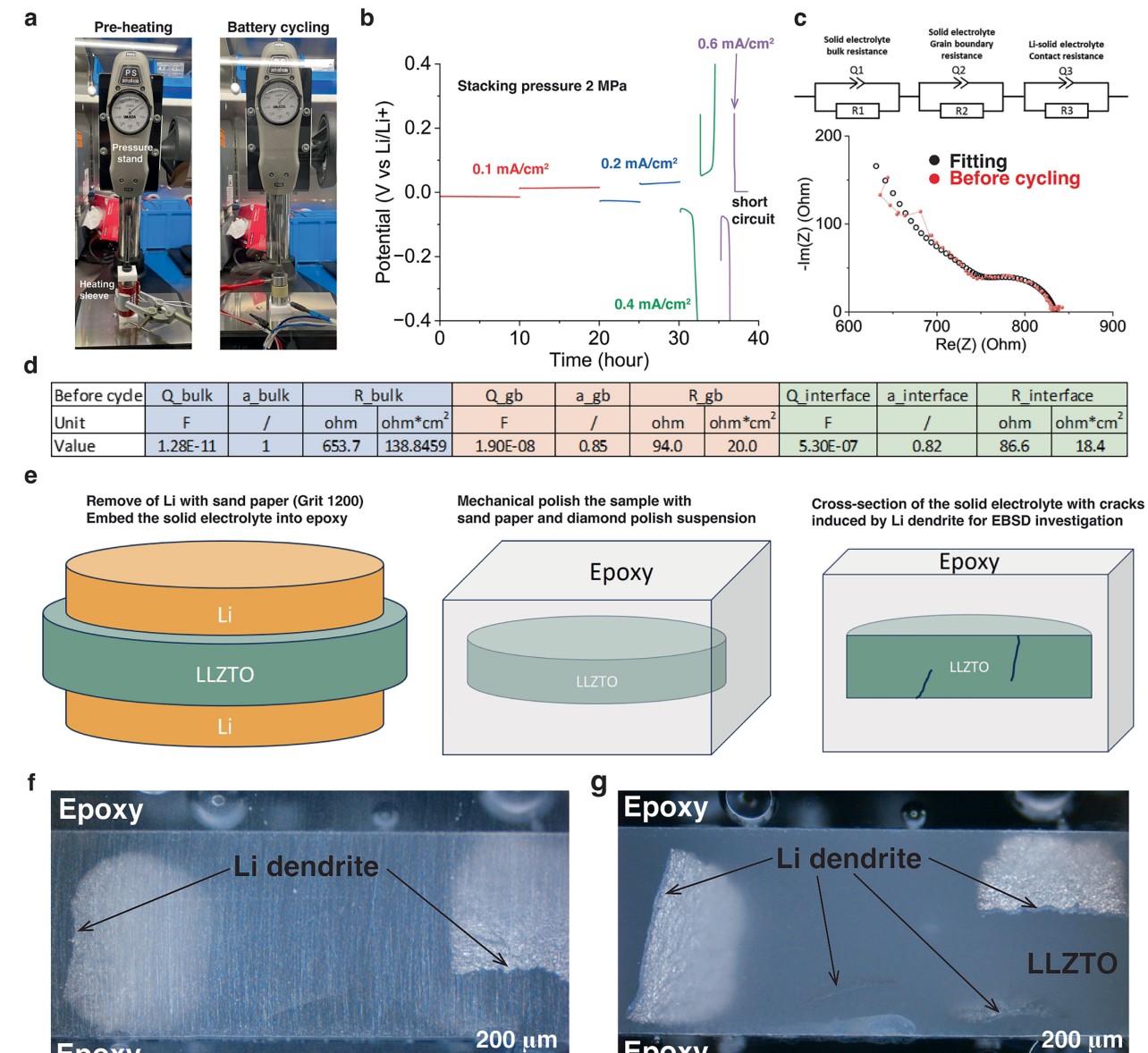

**Extended Data Fig. 1 | Symmetric-cell geometry revealing lithium dendrite penetration. a**, Experimental set-up for electrochemical cycling of the lithium/solid electrolyte/lithium symmetric cell inside an argon-filled glovebox, with controlled stacking pressure and pre-heating to ensure close contact between the lithium metal and the solid electrolyte. **b**, Cell potential response during lithium plating and stripping in a symmetric cell at various current densities, with a fixed areal capacity of 1 mAh cm⁻² per step. **c**, EIS scan before applying bias.

**d**, Fitting parameters for EIS data. **e**, Schematic showing sample preparation of short-circuited solid electrolyte for SEM and EBSD analysis. **f**, Cross-sectional optical microscopy image of a short-circuited solid electrolyte embedded in epoxy. The surface was polished with 320-grit sandpaper. **g**, The same sample after further fine polishing to a greater depth using a SiO₂ polishing suspension. Data source: symmetrical cell.