## [Peer Review File · Nature]

Mechanically Driven Lithium Dendrite Penetration in Garnet Solid Electrolyte

Corresponding Author: Dr Yuwei Zhang

Version 0:

Reviewer comments:

Referee #1

(Remarks to the Author)

In this paper, the authors present comprehensive characterization of lithium filament-driven fracture behavior in oxide solid-state electrolytes for solid-state batteries. A variety of techniques are used to characterize the nature of filament growth, including cryo electron imaging, EBSD, and STEM. The application of these techniques is extremely challenging and novel in this system given the difficulties of working with lithium metal, and the information gained from their combination is quite useful. The main finding that hydrostatic stress in the lithium builds in the growing lithium to cause tensile stress and fracture of the electrolyte is not entirely new (as this theory has been proposed in prior work as well and is a key theory within thin community), but it is substantially advanced through the new evidence provided in the paper. Overall, I am impressed by the characterization in this paper and it is very well done. However, there are some limitations to the results, and I am unsure whether the novelty of the science is high enough for Nature. Please see my comments below:

A broad comment is that, out of necessity and similar to other related papers in this area (e.g., Nature 68, 287-293, 2023), the observations in this paper come from a select few experiments on idealized samples that are different in some ways than a real battery. For instance, there is a metal probe tip that causes current concentrations, and the overpotentials are relatively high (Fig. 1 and S1). I understand this is hard to get around, but the language in the paper makes conclusions that state that the observations and mechanisms proposed are ALWAYS in play. For instance, on page 10, it is worded that the one observation of the single filament means that this mechanism always holds. What about at different current, voltages, electrolyte properties? Indeed I fully agree and expect that the mechanics-driven cracking of SEs is usually why they fail, especially under the usual operating conditions of solid state batteries. But, can Li never nucleate within and grow in various SSEs due to electronic properties? What conditions may allow for this (or not?). I suggest considering of the conditions that may give rise to other possible conclusions, if the authors warrant.

Another broad comment regarding novelty – the overall theory in the literature of fracture-driven filament propagation is bolstered by the evidence here (the characterization is novel, as I've stated), but this theory has been widely explored and supported in other papers as well (e.g., Fincher et al, Joule, Volume 6, Issue 12, 2794 – 2809; 10.1016/j.matt.2023.10.014, 10.1007/s11340-024-01085-7, Nature 68, 287-293, 2023). While the evidence here is very high quality, because of the nature of isolated observations (as described above) I don't think it rules out other possible modes of growth. Thus, while I am certainly impressed and I appreciate the work and believe it is high impact, it is difficult for me to judge whether it is appropriate for Nature.

The experiments involve growing a filament along a free surface. How does this impact behavior compared to growing in the interior of a SE?

The description of the Vickers indents used to represent defects or voids is a bit confusing. Why would the residual indents be considered voids? Voids within a SE would seemingly not have the same deformation and residual stresses surrounding them as indents would. The relationship between intrinsic defects and these induced defects is not clear.

Regarding the implication on page 15 and 16 that the findings “offer design principles for implementing defect based

strategies to control dendrite propagation...” – it is not clear that this is the case. In particular, to compete on specific energy/energy density, SE separators must be thin, 20-30 microns. Dendrite propagation in such thin membranes would be very fast. How could the knowledge gained here be applied to realistic membranes?

Referee #2

(Remarks to the Author)

This work reports the fracture process in garnet electrolytes driven by lithium dendrites using multiscale cryo S/TEM and modelling, to understand the long-standing debate on “soft-penetrates-hard” failure. They revealed that hydrostatic stress buildup within the confined lithium is the main reason for lithium penetration in solid electrolyte, rather than electron leakage or isolated lithium nucleation ahead of the tip. The authors proposed a strategy using geometrically engineered elliptical voids within the electrolyte to redirect dendrite growth paths and prevent short-circuiting. The results are interesting. However, this reviewer feels that the data presented are not convincingly enough to justify their correctness at this stage. And, the chemical environment or surface free energy of the solid electrolyte differs from its bulk properties, which may induce distinct dendrite growth mechanisms at the surface compared to the interior. The authors' in-plane battery configuration cannot accurately represent real conditions. Therefore, this manuscript is not recommended for publication in Nature. Additional comments are as follows:

1. The authors used phase-field simulations to predict 600 MPa hydrostatic stress in Li, but no direct experimental measurement is provided.
2. Page 3, “...provide indirect, greyscale-based imaging of lithium and cracks”. This is an unfair claim. OM, X-ray, SEM are direct observations as the work presented here. The difference is resolution and analysis ability.
3. The in-plane cell geometry applied here could not represent the other geometries. How the observations here are representative is questionable. Moreover, a 2D slab solid electrolyte model is demonstrated here, however, the 3D lithium growth in real electrolytes could be different.
4. The confirmation of the presence of lithium in the crack was done by STEM-EELS of the FIB lamella samples. As already been known that lithium metal could be reduced by ion beam or electron beam bombardment. Therefore, complicated reactions could occur during the FIB processing. This may not be a suitable way to reveal the original status of the matter inside the crack. Direct detection of lithium in the thicker SEM samples is more reliable. Moreover, raw data of the EELS spectra is suggested. MSA analysis may bring artifacts and details of the data processing was not provided.
5. Again, the conclusion related to lattice distortion or rotation relies on the analysis of the FIB samples. Ion milling may cause the release or rearrangement of the strain. How the results are representative for the real situation is doubtful.
6. The authors employed optical microscopy to study the top-view growth direction of lithium dendrites near indent arrays, but this method has limitations. Since lithium dendrites can propagate in 3D within the solid electrolyte, a 2D top-view observation cannot confirm whether the dendrites actually intersect with the indents in the depth direction. Additionally, the resolution of optical microscopy is too low to definitively determine whether the observed features are indeed lithium dendrites or cracks, as claimed by the authors.
7. The authors artificially created voids using Vickers indents and observed that the growth direction of lithium dendrites deflected upon reaching the indent array. However, their conclusion seems overly speculative. An indent is not simply a void—it primarily consists of the indentation core and the surrounding deformation zone. The indentation core is the region directly formed by the indenter's action, with its shape and dimensions closely related to the indenter's geometry and loading conditions. The surrounding deformation zone arises from plastic flow and stress transfer in the material during the indentation process. Therefore, the possibility cannot be ruled out that the deflection of lithium dendrites is caused by their interaction with the deformation zone around the indent rather than by the voids alone.
8. The authors suggest that artificially introduced voids can alter dendrite growth direction and prevent short circuits, but such voids may simultaneously reduce the ionic conductivity of the solid electrolyte. Additionally, the authors propose that “interfaces in multilayer solid electrolytes may serve as natural voids”. What is the underlying rationale for this claim?

While I appreciate the authors' efforts to investigate the origin of dendrite penetration, the findings remain largely phenomenological. The observed results may represent either strain-relaxed states or artifacts induced by ion beam modification, rather than capturing the material's original state. Consequently, the claim of uncovering the ‘origin’ appears overstated. Overall, this work offers only incremental advancement compared to prior studies in the field.

Version 1:

Reviewer comments:

Referee #1

(Remarks to the Author)

Both reviewer comments had a number of similarities, including regarding how representative the experimental geometry is and differences between the indents and true void structure/stress state. The authors have carried out a robust and extensive series of additional experiments to address the variety of comments. I appreciate the additional experiments in the symmetric cell with different geometry, and in general the prior and new experimental results are interesting.

The experimental findings in this paper provide new understanding of lithium growth causing stress and fracture, and I think they are useful for the community. The experiments are difficult and the results are high quality. Given the positive but

somewhat reserved response from both reviewers, I think the paper could be acceptable in Nature if the editors are sufficiently excited about the paper.

I do have one additional comment. My original comment on real solid-state separators being thin was not addressed in accord with how it was originally intended. My point was not necessarily that filaments would grow quickly through thin (20-30 micron) solid state separators, but instead that with such thin separators there is much less volume over which to carry out any sort of pore or void engineering strategy. In a 1-mm thick pellet, voids can be engineered easily throughout this thickness. but in a 20-micron film, void creation might affect the entire cross section of the film. I believe the authors should address this question in the text: is void engineering realistic in 20 micron films that need to be produced over meters-square area and must be highly uniform and ionically conductive? This is not addressed in the revision.

Referee #2

(Remarks to the Author)

The additional experimental results have improved the revised manuscript compared to the original version. After reviewing all materials, I have some major concerns:

1. Both I and the other reviewer raised concerns regarding whether the specially designed 2D in-plane geometry setup accurately represents real 3D scenarios. I anticipate future readers sharing this concern. While the authors' revisions primarily addressed this point, the core structure remains unchanged: the main text still focuses on 2D results, relegating all 3D results to the SI. The minor revisions to the main text are insufficient to alleviate potential reader doubts. At minimum, key 3D results should be included in the main figures.
2. The EIS data presented in Figure S5c and S9c for the symmetric Li/LLZTO/Li cells raise significant concerns regarding data quality and interpretation. The Nyquist plots exhibit an unusual, highly compressed semicircular shape in the high-to-medium frequency range. The manuscript presents the raw EIS data without accompanying equivalent circuit fits or any quantitative analysis.
3. The manuscript completely lacks of any description or discussion related to Figure S9 in the main text. Figure S9 presents critical electrochemical data for the symmetric cell. These data are fundamental for establishing the operational state and stability of the symmetric cell. Specially, the stripping voltage profile presented in Figure S9b is highly abnormal for a symmetric Li/LLZTO/Li cell and raises serious concerns regarding the interpretation of the experiment.
4. The practicality and novelty of void strategy remain questionable, as raised by Reviewer 1. The discussion on page 18 acknowledges the trade-off with ionic conductivity but remains speculative. The suggestion that bilayer electrolyte interfaces could serve as "natural voids" is interesting but underdeveloped. The void introduction, conductivity loss, and scalability must be solved for practical application.
5. The authors have conducted extensive new analyses, including residual stress measurements to demonstrate that the deformation zone surrounding the Vickers indents has negligible influence on Li dendrite deflection. An indent is a complex mechanical feature consisting of a plastically deformed core and a surrounding strain-affected zone, which is not merely a geometric void. The authors rely on phase-field simulations that treat the indent as a void, but this model does not explicitly incorporate the possible effects of the deformed material surrounding the indent.
6. My concern, echoed by the other reviewer, persists. While this work provides a solid characterization of lithium dendrite penetration through cracks (which I appreciate), it cannot definitively exclude other mechanisms. Real battery scenarios likely vary. Consequently, the manuscript's conclusion and title ("mechanical origin of...") appear overstated; this mechanism is likely only one possible origin. Furthermore, the proposed surface-void-engineering strategy seems impractical, facing significant trade-offs with ionic conductivity and structural integrity/stability, as acknowledged by the authors in their response. Therefore, I strongly recommend toning down the conclusions. However, if weakened, I question whether the manuscript's significance meets Nature's standards relative to other publications in the field.

Version 2:

Reviewer comments:

Referee #1

(Remarks to the Author)

I have examined the revised paper and the authors' response to reviewer comments. I appreciate the additions related to uncertainty and challenges of pore or bilayer engineering to control dendrite growth. I believe the authors have improved the paper throughout this revision process, and that the paper contains useful and important information for the field. If the editors are sufficiently excited, I believe it could be accepted.

Referees' comments:

Referee #1 (Remarks to the Author):

In this paper, the authors present comprehensive characterization of lithium filament-driven fracture behavior in oxide solid-state electrolytes for solid-state batteries. A variety of techniques are used to characterize the nature of filament growth, including cryo electron imaging, EBSD, and STEM. The application of these techniques is extremely challenging and novel in this system given the difficulties of working with lithium metal, and the information gained from their combination is quite useful. The main finding that hydrostatic stress in the lithium builds in the growing lithium to cause tensile stress and fracture of the electrolyte is not entirely new (as this theory has been proposed in prior work as well and is a key theory within this community), but it is substantially advanced through the new evidence provided in the paper. Overall, I am impressed by the characterization in this paper and it is very well done. However, there are some limitations to the results, and I am unsure whether the novelty of the science is high enough for Nature. Please see my comments below:

Authors' response: We sincerely thank the reviewer for his/her compliments and constructive comments. We also thank the reviewer for acknowledging the extreme difficulty of the experiments.

To address the relevance of the in-plane experiments here to real battery configurations, we have now performed a full set of new experiments using a Li/LLZTO/Li symmetric cell configuration. Specifically, we repeated all cryogenic characterizations on symmetric cells, including cryogenic EBSD characterization of fractured LLZTO (Fig. S5-S6), three-dimensional fracture reconstruction (Fig. S9-S10), TKD characterization of the Li dendrite (Fig. S17) and 3D micromechanical fracture modelling (Fig. S36). These new measurements reproduce the key mechanistic features observed in the in-plane cell, demonstrating that our conclusions are not specific to the geometry but are intrinsic and general features of lithium-driven fracture in garnet LLZTO solid electrolyte. The complete symmetric-cell dataset is provided in the Supplementary Information.

We also quantified the residual stresses induced by the Vickers indent and assessed their potential influence on crack deflection. To do this, we compared the Raman peak shift (Fig. S32) near the indent with that in the undeformed region and employed a multibeam optical stress sensor (Fig. S33) to quantify the residual stress around the Vickers indent. All measurements consistently show a residual stress of approximately 1 MPa, which our fracture modelling (Fig. S33) indicates that this level of residual stress has a negligible influence on the observed crack deflection.

1 We believe that addressing these points has further improved the clarity and overall novelty of
2 the manuscript. The perceived limitations are clarified, and the novelty of the science is
3 highlighted. Detailed responses to the reviewer's specific comments are provided below

1 Referee 1 comment 1: A broad comment is that, out of necessity and similar to other related
papers in this area (e.g., Nature 68, 287-293, 2023), the observations in this paper come from a
select few experiments on idealized samples that are different in some ways than a real battery.
For instance, there is a metal probe tip that causes current concentrations, and the overpotentials
are relatively high (Fig. 1 and S1). I understand this is hard to get around, but the language in the
paper makes conclusions that state that the observations and mechanisms proposed are ALWAYS
in play. For instance, on page 10, it is worded that the one observation of the single filament
means that this mechanism always holds. What about at different current, voltages, electrolyte
properties? Indeed I fully agree and expect that the mechanics-driven cracking of SEs is usually
why they fail, especially under the usual operating conditions of solid state batteries. But, can Li
never nucleate within and grow in various SSEs due to electronic properties? What conditions
may allow for this (or not?). I suggest considering of the conditions that may give rise to other
possible conclusions, if the authors warrant.

**Authors' response:**

To address the concern that the *in-plane battery configuration differs from a real battery setup*,
we fully comply with this concern and we have therefore now performed a full additional suite of
symmetrical cell experiments, including cryogenic EBSD characterization of fractured LLZTO (Fig.
S5-S6), 3D fracture reconstruction (Fig. S9-S10), TKD characterization of the Li dendrite (Fig. S17)
and 3D micromechanical fracture modelling (Fig. S36). These new results are now fully
incorporated into the Supporting Information. Compared to the results performed through
symmetrical cell, the in-plane geometry allows unambiguous identification of the dendrite tip (Fig.
2), and facilitates site-specific cryogenic characterization at the dendrite tip.

- • Fig. S5-S6 show the fracture pattern of the solid electrolyte induced by the growth of Li
dendrite in the symmetrical cell and allow direct comparison with the in-plane cell
results in Fig. 1. Fig. S6d-S6f show the crystallographic lattice orientation around the
fractured solid electrolyte. Consistent with the in-plane configuration, we again observe
both intergranular and transgranular fracture modes of the solid electrolyte induced by
Li dendrite propagation in the symmetrical cell, with approximately 20-30% of the
fracture events being transgranular. However, it is not possible to accurately quantify the
intergranular-transgranular ratio or determine the crack-deflection angle because the
fracture contour is not well-defined in the symmetric-cell geometry, in contrast to the
clearly resolved crack paths shown in Fig. 1e.
- • Fig. S9-S10 present the 3D fractography of the solid electrolyte in the symmetrical cell and
enable direct comparison with the in-plane cell results in Fig. 2. The reconstructed Li
dendrite morphology in Fig. S10 exhibits a tortuous geometry similar to that shown in Fig.
2b in the main manuscript. Notably, the seemingly isolated Li dendrite #1-#3 observed in
Fig. S10e (slice 350) were revealed through the full 3D reconstruction to originate from a

continuous dendrite that is already visible in slice 1 (Fig. S10e). Without 3D reconstruction
at cryogenic temperature, such an observation could be misinterpreted as evidence for
isolated Li nucleation and propagation within the LLZTO electrolyte.

- • Fig. S17 demonstrates the microstructure of Li dendrites in the symmetric cell and allows
direct comparison with the in-plane cell results in Fig. 3. Figs. S17c and S17d present the
lattice orientation map and grain reference orientation deviation (GROD) angle map of the
Li dendrite, respectively. The Li dendrite grain size is of the same order of magnitude as
that observed in Fig. 3. The GROD map in Fig. S17d reveals minimal lattice rotation across
most of the dendrite volume. Modest orientation gradients are observed only near the
lithium/LLZTO interface, whereas the dendrite interior exhibits negligible gradients,
indicating the absence of significant plastic strain. All these observations are consistent
with those in Fig. 3 and further support our conclusion that hydrostatic stress, rather than
plastic deformation, serves as the dominant driving force responsible for the “soft-
penetrates-hard” behavior during lithium dendrite growth.
- • Fig. S36 shows the 3D micromechanical fracture modeling results for the symmetric cell
configuration and allows direct comparison with the 2D modelling results in Fig. 3. The
hydrostatic stress buildup within the Li dendrite remains orders of magnitude higher than
the von Mises stress, consistent with the 2D simulation results. These 3D modelling results
therefore reinforce our conclusion that the hydrostatic stress contribution within the Li
dendrite dominates the mechanical driving force for the dendrite-induced fracture of the
LLZTO electrolyte.

All these results from Figs. S5-S6, S9-S10, S17 and S36 together demonstrate that the Li dendrite
formation behavior observed in the in-plane cell is consistent with that occurring in the
symmetrical cell.

Regarding the comment that *“the observations and mechanisms proposed are ALWAYS in play”*,
we have revised statements in the manuscript that could be interpreted as implying universal
applicability, acknowledging that our conclusions are specific to the model Li/LLZTO cell system
studied here.

Regarding the comment that *“But, can Li never nucleate within and grow in various SSEs due to
electronic properties? What conditions may allow for this”*, we fully agree that lithium nucleation
due to electronic leakage cannot be universally excluded across all solid electrolytes. However, in
our experiments, both in the main text and in the new additional dataset (Fig. S14), we find no
evidence for isolated Li nuclei ahead of the crack tip in LLZTO. To address this point more explicitly,
we performed new well-targeted experiments:

• We electroplated a Li dendrite using the same current as in Fig. 1. The overpotential during
dendrite growth was comparable to that in Fig. 1, approximately 50 mV vs Li/Li⁺. An
additional TEM lamella was lifted out from a region approximately 5 μm ahead of the Li
dendrite prepared through in-plane configuration, following the same procedure as in Fig.
2e. If triple junctions promoted formation of isolated Li nuclei during electrochemical
cycling, Li EELS signals would be expected in these regions. However, no EELS signal
corresponding to Li was detected within the triple junctions (Fig. S14). These observations
indicate that, within the typical operating voltage window of solid-state batteries¹ (< 4.5
V vs Li/Li⁺), no isolated Li nuclei are detected ahead of the crack tip, suggesting that
electronically driven Li nucleation does not occur under these conditions in LLZTO. Liu *et*
*al.*² (Nature Materials 20.11 (2021): 1485-1490.) reported the formation of Li metal
deposits at triple junctions during in situ TEM cycling (Fig. 3 in Liu *et al.*²). As stated in their
study, Li nucleation was only observed in the triple junction under a high applied bias of
10 V vs Li/Li⁺, while no Li formation occurred at 2 V or 5 V vs Li/Li⁺. However, such a high
voltage (10 V) is far beyond usual operating conditions of solid-state batteries¹ (< 4.5 V vs
Li/Li⁺). Our observations in Fig. S14 are therefore consistent with those of Liu *et al.*² under
realistic operating conditions, in which no isolated Li nucleation is detected.

**Modification to the manuscript:**

Page 6: This configuration promotes the formation of a single, macroscopically straight, through-
thickness dendrite initiated from a tungsten needle, under an overpotential of ~50 mV vs Li/Li⁺
during electrochemical biasing (Fig. 1c).

Page 7: The lithium-plating-induced fracture behavior of the Li/LLZTO/Li symmetrical cell was
 also examined, as shown in Figs. S5 and S6. Both intergranular and transgranular fractures features
 were observed, with approximately 20-30% of the fracture events being transgranular. However,
 it is not possible to accurately quantify the intergranular-transgranular ratio or determine the crack-
 deflection angle because the fracture contour is not well-defined in the symmetric-cell geometry,
 in contrast to the clearly resolved crack paths shown in Fig. 1e.

 Figure S5. (a) Experimental setup for electrochemical cycling of the lithium/solid electrolyte/lithium
 symmetric cell inside a glovebox, with controlled stacking pressure and pre-heating to ensure close contact
 between the lithium metal and the solid electrolyte. (b) Cell potential response during lithium plating and
 stripping in a symmetric cell at various current densities, with a fixed areal capacity of 1 mAh cm⁻² per step.
 (c) EIS scan before applying bias. (d) Schematic showing sample preparation of shorted solid electrolyte
 for SEM and EBSD analysis. (e) Cross-sectional optical microscopy image of a short-circuited solid
 electrolyte embedded in epoxy. The surface was polished with 320-grit sandpaper. (f) The same sample
 after further fine polishing to a greater depth using a SiO₂ polishing suspension. Data source: symmetrical
 cell.

Figure S6. (a) Optical microscopy image highlighting the region selected for cryogenic SEM and EBSD
characterization. (b, c) SEM images of a short-circuited LLZTO solid electrolyte prepared under a
symmetric cell configuration. (d-f) EBSD maps showing Li plating-induced fracture in the LLZTO solid
electrolyte under the same configuration. Data source: symmetrical cell.

**Page 9: The three-dimensional fractography of LLZTO in the Li/LLZTO/Li symmetric cell was**
**further analyzed, as presented in Figs. S9 and S10. The reconstructed lithium dendrite morphology**
**in Fig. S10 exhibits a tortuous geometry similar to that shown in Fig. 2b. Notably, the seemingly**
**isolated lithium dendrite #1 to #3 (Fig. S10e, slice 350) originate from a single continuous dendrite**
**(Fig. S10e, slice 1). Without three-dimensional reconstruction at cryogenic temperature, such an**
**observation could be misinterpreted as evidence for isolated lithium nucleation and propagation**
**within the LLZTO electrolyte.**

Figure S9. (a) Potential response during lithium plating and stripping in the symmetric cell at 0.6 mA cm^{-2} .
The cell was shut down after 600 s during the stripping stage. (b) Enlarged view of the stripping period. (c)
Electrochemical impedance spectroscopy results before and after cycling. Data source: symmetrical cell.

 Figure S10. (a) Top-view optical microscopy image of the cycled solid electrolyte corresponding to the
 cycling curve in Figure S9a. The red circle marks the region selected for electron microscopy
 characterization. (b) Magnified view of the cracked region used for subsequent analysis. (c, d) Three-
 dimensional reconstruction of Li dendrites in the solid electrolyte grown in the symmetric cell configuration,
 displayed from two different viewing angles. (e) Selected cross-sectional slices used for the 3D
 reconstruction shown in (c) and (d). Labels #1-#3 mark three seemingly isolated cracks in Slice 350, which
 originally branched from a single major crack, as shown in Slice 1 and Fig S10c. Data source: symmetrical
 cell.

 **Page 13: The microstructure of the lithium dendrite growth within the solid electrolyte under the**
 **symmetric-cell configuration is shown in Fig. S17. The GROD map also exhibits minimal lattice**
 **rotation, comparable to that observed in Fig. 3c.**

 Figure S17. (a) Cross-sectional view of a lithium dendrite penetrating the solid electrolyte after
 electrochemical cycling in the symmetric cell configuration. The red square marks the region selected for
 TKD measurement. (b) Side view of the lift-out lamella prepared for TKD analysis, showing that the lithium
 dendrite exhibits a tortuous three-dimensional morphology. (c) TKD-SEM map showing the grain
 orientation. (d) GROD map showing the grain orientation.

orientations of the lithium dendrite. (d) Grain reference orientation deviation map of the same region. Data
source: symmetrical cell.

Page 25: These three-dimensional simulations (Fig. S36) verify the two-dimensional results and
confirm that the predicted fracture behavior remains consistent across both geometries.

Figure S36. Three-dimensional simulations of the maximum tensile stress in LLZTO ahead of the dendrite
tip, the average von Mises stress within the lithium dendrite and the average hydrostatic stress within the
lithium dendrite. The yield strength of the lithium is assumed to be (a) 12.5 MPa, (b) 62.5 MPa, and (c) 125
11 MPa. In (d), it is assumed that the lithium dendrite undergoes no plastic deformation. The results confirm
that hydrostatic stress within the lithium dendrite, rather than deviatoric stress, is the dominant fracture-
driving force.

Page 10: Previous work by Liu *et al.*² reported that the band gap at the grain boundaries of garnet
solid electrolytes appears to be lower than that of the bulk, enabling lithium nucleation at triple
junctions during *in situ* TEM biasing at 10 V vs. Li/Li⁺, whereas no lithium nuclei were observed
at lower biases of 2 V and 5 V. In Fig. S14, we show an additional TEM lamella prepared ahead of
the dendrite tip. The dendrite was grown under an overpotential of 50 mV vs Li/Li⁺. No EELS
signal corresponding to lithium was detected within the triple junctions (Fig. S14). These results
not only agree with the observations of Liu *et al.*² at lower bias, but also suggest that within the
normal battery operating voltage window¹ (< 4.5V vs Li/Li⁺), lithium plating does not produce

measurable isolated lithium accumulation at grain boundaries or triple junctions in LLZTO
electrolytes.

Figure S14. (a) Cross-sectional observation of a region located approximately 5 μm ahead of the dendrite
tip. The size and geometry of the triple junctions exhibit feature similar to those in pristine LLZTO, as
shown in Figures S8. (b, c) Corresponding EELS lithium count maps of the boxed regions in (a), confirming
the absence of lithium ahead of the dendrite tip. EELS spectra of (d) low loss and (e) Li-K edge. Data source:
in-plane cell.

Page 10: Overall, cryo-SEM and cryo-STEM observations show that lithium fully occupies the
nanoscale crack at the dendrite tip.

**Modification to the Method section:**

Page 20: Electrochemical Impedance Spectroscopy (EIS) data were recorded in the frequency
range between 10 Hz and 7 MHz with an amplitude of 50 mV using a SP200 impedance analyzer
(Bio-Logic).

Page 20: Lithium dendrite growth through the symmetric cell configuration was cycled using the
same potentiostat equipped with a pressure stand (Imada Inc.). Before applying the bias, the
symmetric cell was heated to 130 $^{\circ}\text{C}$ using a heating sleeve (RS Components Ltd.) to improve the
interfacial contact between the lithium metal and the solid electrolyte. After short-circuiting, the

lithium metal was removed using sandpaper with a grit size of 1200. The short-circuited solid
electrolyte was then taken out of the glovebox, soaked in epoxy overnight for curing, and
subsequently polished to the region where features resembling lithium dendrites could be observed,
as shown in Figs. S5e and S5f. Fig. S5d schematically illustrates the sample preparation procedure.
Fig. S5e shows the surface after rough polishing with 320-grit sandpaper, while Fig. S5f shows the
result after fine polishing using a 0.1 μm SiO_2 polishing suspension. The lithium metal on the
plating side, where dendrite growth occurred, could be easily peeled off by hand in the results
shown in Figs. S9, S10 and S17. Therefore, no sandpaper was used to remove the lithium electrode,
in contrast to the procedure used for the samples shown in Figs. S5–S6.

Referee 1 comment 2: Another broad comment regarding novelty – the overall theory in the
literature of fracture-driven filament propagation is bolstered by the evidence here (the
characterization is novel, as I've stated), but this theory has been widely explored and supported
in other papers as well (e.g., Fincher et al, Joule, Volume 6, Issue 12, 2794 – 2809;
10.1016/j.matt.2023.10.014, 10.1007/s11340-024-01085-7, Nature 68, 287-293, 2023). While
the evidence here is very high quality, because of the nature of isolated observations (as
described above) I don't think it rules out other possible modes of growth. Thus, while I am
certainly impressed and I appreciate the work and believe it is high impact, it is difficult for me to
judge whether it is appropriate for Nature.

**Authors' response:**

We appreciate the reviewer's positive feedback. Our understanding is summarized below:

**Previous studies:** For LLZO, *Fincher et al.*³ (Joule, Volume 6, Issue 12, 2794 – 2809) provided
evidence that the application of external mechanical load can control the growth of Li dendrites.
*Athanasίου et al.*⁴ (Matter 7.1 (2024): 95-106) observed substantial stress associated with
dendrite propagation. For LPSC, *Ning et al.*⁵ (Nature 618.7964 (2023): 287-293) showed that the
onset of a short circuit occurred only well after cracks propagated fully through the solid
electrolyte. Collectively, these studies support a mechanically driven fracture mechanism for Li
dendrite propagation, but primarily through macroscopic-scale characterization methods such as
X-ray and optical microscopy imaging. Importantly, none of these works directly examined
whether isolated Li nuclei form ahead of the dendrite tip, nor did they probe the nanoscale Li
dendrite-solid electrolyte interaction region where fracture initiates. This information is critical
for distinguishing between the existing understandings, which lead to two almost orthogonal
design directions.

**Our contribution:** Our work extends current understanding by providing nanoscale evidence of
the Li dendrite–solid electrolyte interaction region. Moreover, our statistical analysis of
transgranular and intergranular fracture behavior provides a quantitative framework for
estimating the grain-boundary fracture toughness, a critical parameter for designing dendrite-
free solid electrolytes. Detailed contributions are explained below:

- 1. We show, for the first time with cryogenic STEM/TEM and EBSD, that the Li dendrite fully
fills the nanoscale crack within the LLZTO electrolyte, addressing a long-standing debate.
Furthermore, as discussed in our response to Comment 1, we observe no isolated Li nuclei
ahead of the crack tip within the normal operating voltage window of solid-state batteries.
Even acknowledging reports of Li nucleation at triple junctions² under extreme bias (10 V
vs Li/Li⁺), our results demonstrate that under realistic conditions the hydrostatic pressure
generated inside the lithium dendrite is sufficiently high to fracture the solid electrolyte,
without requiring electronically driven Li nucleation.

- 2. Our work provides the first statistical evaluation showing that the grain-boundary fracture
toughness of the solid electrolyte is a factor of 3–5 lower than that of the bulk, based on
micromechanical modelling and cryogenic EBSD analysis. This parameter is extremely
challenging to measure or compute directly. Our approach (Fig. 1h) introduces a new
framework for battery materials engineering, emphasizing grain-boundary mechanical
strengthening as a design strategy for dendrite suppression.
- 3. We provide direct characterization of the microstructure of Li dendrites. Through TKD and
GROD analyses, we reveal that Li dendrites exhibit minimal lattice rotation and negligible
plastic strain. When combined with our 3D micromechanical fracture modelling, these
results demonstrate that hydrostatic pressure buildup in the lithium, rather than von
Mises stress, is the dominant driving force behind the brittle fracture of garnet solid
electrolytes. This clarifies the mechanical origin of the “soft-penetrates-hard” behavior
during Li dendrite growth.

**Modification to the manuscript:**

We note that part of this comment requires the same revision as addressed in *Referee 1*
*comment 1*. To avoid redundancy, we have implemented the corresponding modification on
*Page10, line 5*. We therefore do not repeat the contents here.

**Page 13: The simulated stress distribution ahead of the propagating lithium dendrite matches well**
**with the experimental measurements reported by Athanasiou-Fincher *et al.*⁴**

Referee 1 comment 3: The experiments involve growing a filament along a free surface. How does
this impact behavior compared to growing in the interior of a SE?

**Authors' response:**

We clarify that in the in-plane cell configuration, the dendrite grows within the thin LLZTO
electrolyte (thickness approximately 150 μm), as shown in Fig. S11, rather than along the surface
of the LLZTO electrolyte. This geometry promotes the formation of a single, macroscopically
straight, through-thickness dendrite initiated at the tungsten needle under an overpotential of
~ 50 mV vs Li/Li⁺ during electrochemical biasing (Fig. 1c). In addition, we have performed further
experimental and simulation studies on the lithium/LLZTO/lithium symmetric cell, as detailed in
our response to *Referee 1, Comment 1*.

**Modification to the manuscript:**

We note that this comment requires the same revision as addressed in *Referee 1 comment 1*. To
avoid redundancy, we have implemented the corresponding modification on *Page 7 line 1, Page*
*9 line 6, page 13 line 11 and Page 25 line 6*. We therefore do not repeat the contents here.

Referee 1 comment 4: The description of the Vickers indents used to represent defects or voids
is a bit confusing. Why would the residual indents be considered voids? Voids within a SE would
seemingly not have the same deformation and residual stresses surrounding them as indents
would. The relationship between intrinsic defects and these induced defects is not clear.

**Authors' response:**

We intended the cracks generated by the indents to be considered as transverse voids, rather
than the residual indents themselves. We thank the reviewer for bringing this point up. High-
resolution images of the interaction between Li dendrite and sharp cracks induced by Vickers
indent are added as Fig. 4. Related to the deformation and residual stress fields that surround the
indents, we agree with the reviewer that the stress field can also influence dendrite propagation,
as reported by Fincher *et al.*³, who showed that stresses on the order of hundreds of MPa can
lead to Li-dendrite deflection in garnet solid electrolytes. However, in our experiments we
observed cracks around the Vickers indents, which are expected to relax the residual stress. This
interpretation is supported by our residual stress measurements, which show less than 1 MPa of
stress remaining around the indent. In response to a separate reviewer comment (*Referee 2,*
*Comment 7*), our detailed explanation is provided below:

- • To estimate the residual stress induced by plastic carriers, we performed cryo-EBSD
mapping around the Vickers indent. If plastic flow had occurred around the indent,
variations in the kernel average misorientation (KAM) map (proportional to geometrical
necessary dislocation density) should have been observed. However, the results in Fig.
S31b-c show no discernible trend or increase in KAM values. Furthermore, the sample
surface was mechanically polished (down to a 0.1 μm SiO_2 suspension) to remove
approximately 20 μm of the surface layer and to investigate potential subsurface residual
stress induced by plastic flow. The subsurface results (Fig. S31e-f) were consistent with
those from the surface region, showing no measurable variation in KAM values around
the Vickers indent.
 - • To estimate the residual stress induced by elastic response, we first compared the Raman
spectra obtained near the Vickers indent—where residual stress may exist—with those
from the undeformed region of the garnet solid electrolyte. As shown in Fig. S32, no
measurable peak shift was observed between the region close to the Vickers indent and
the undeformed region. Second, we employed a more sensitive technique—multibeam
optical stress sensor—which can detect a change in radius of curvature on the order of 20
34 km, corresponding to a strain variation of approximately 0.01% in a 0.5 mm-thick LLZTO
solid electrolyte. Fig. S33a shown the schematic of experimental setup and Fig. S33b show
the curvature change before and after inducing a 7*4 array Vickers indent (spacing 150
37 μm) into the solid electrolyte. Plugging the curvature change into Stoney equation, we get

a residual compression around 1MPa. By incorporating this value into our phase field
modelling (Fig. S33d and S33e), we found the measured residual stress won't influence
the crack propagation path in our work.

As a summary, deflection of lithium dendrites is indeed caused by the interaction with the cracks
induced by the Vickers indent. The deformation zone plays a negligible role on the interaction of
Li dendrite with the cracks.

**Modification to the manuscript:**

Page 16: The post-mortem cryogenic SEM image (Fig. 4d) shows that the lithium dendrite
propagates toward the cracks emanating from the Vickers indent and subsequently deflects by
approximately 45°, following the pre-existing crack induced by Vickers indent. The subsurface
interaction between the lithium dendrite and the LLZTO solid electrolyte is shown in Fig. S29.

Figure S29. (a) Deflection of a lithium dendrite after interacting with Vickers indents. (b) Cross-sectional
fractography of LLZTO induced by lithium plating in the region highlighted by the red square in (a). The
region of interest was intentionally thinned down to ~ 100 nm to demonstrate that it is fully filled; otherwise,
the background would be visible through the dry crack. Data source: in-plane cell.

**Fig. 4. Tailor lithium dendrite growth through engineered voids.** (a, b) Top-view optical microscope
image showing lithium dendrite growth (a) before and (b) after encountering arrays of Vickers indents. The
dendrite deflects upon reaching cracks emitting from the Vickers indent, altering its trajectory. (c) Zoom-in
optical microscope image of a representative Vickers indent. (d) Zoom-in SEM image showing the
interaction between the crack network and the lithium dendrite. (e-h) Phase-field simulation of lithium
dendrite interaction with (e) circular and (f) transverse voids embedded in the solid electrolyte, showing
hydrostatic stress in lithium and maximum tensile stress in LLZTO during dendrite propagation. Transverse
voids promote deflection, while circular voids allow straight propagation. (i) Schematic of mechanically
driven fracture during lithium dendrite propagation, leading to failure in solid-state batteries. (j) Schematic
of the stress field in the lithium dendrite and solid electrolyte, with hydrostatic pressure buildup in lithium
dendrite and tensile stress in the surrounding solid electrolyte. CAM: cathode active material; SE: solid
electrolyte.

Page 16: We note that the residual stress in polycrystalline LLZTO is substantially relaxed by
 cracking, and our measurements indicate that only ~ 1 MPa remains. Previous studies³ have shown
 that dendrite deflection in garnet electrolyte requires stresses of the order of 100 MPa, whereas the
 residual stress surrounding our Vickers indents is two orders of magnitude lower. Accordingly, our
 fracture modelling indicates that this level of stress has a negligible influence on the observed
 crack deflection. Detailed measurements and calculations are provided in Figs. S31-S33.

Figure S31. Kernel average misorientation (KAM) maps. (a) Schematic showing the sample preparation
 workflow for KAM mapping. (b-d) KAM maps of the regions around the Vickers indent. (e-g) KAM maps
 of the subsurface regions beneath the Vickers indent.

 Figure S32. (a) Raman spectra collected from the region near the Vickers indent and from an undeformed
 reference area. Only two sets of data are shown, and the spectra are vertically shifted for clarity. (b) Optical
 micrograph showing the locations where the Raman measurements were performed. The red circles
 represent the scan near the Vickers indent, and the blue circles represent the scan in the undeformed region.

 Figure S33. (a) Schematic of the multibeam optical stress sensor (MOSS) used to measure the residual
 stress induced by Vickers indentation. (b) Curvature changes of the polycrystalline LLZTO solid
 electrolyte before and after producing an array of Vickers indents (7×4 array with $150 \mu\text{m}$ spacing and 1
 11 kg load). (c) Curvature changes of polycrystalline Zn before and after producing an array of Vickers
 indents (5×4 array with $200 \mu\text{m}$ spacing and 500 g load). This comparison highlights the markedly
 different residual stress levels typically found in brittle ceramics and ductile metals and validates the
 reliability of the MOSS method for capturing stress-induced curvature changes in both material classes.
 (d, e) Simulated lithium dendrite propagation paths under a residual compressive stress of 1 MPa,
 initiated from a vertical and an inclined notch, respectively.

Referee 1 comment 5: Regarding the implication on page 15 and 16 that the findings “offer
design principles for implementing defect-based strategies to control dendrite propagation...” –
it is not clear that this is the case. In particular, to compete on specific energy/energy density, SE
separators must be thin, 20-30 microns. Dendrite propagation in such thin membranes would
be very fast. How could the knowledge gained here be applied to realistic membranes?

**Authors' response:**

We thank the reviewer for this comment. In ceramic solid electrolytes, the flaw-sensitive crack
length is typically on the order of 10 nm or less.⁶ Once the applied stress intensity factor reaches
the fracture toughness, any pre-existing flaw, whose characteristic length scale is several orders
of magnitude smaller than the thickness of even a very thin membrane, can act as the initiation
site for crack propagation. This implies that, although thin separators (20-30 μm) indeed enable
rapid dendrite penetration, the fundamental physics of fracture initiation and crack deflection
remain the same as those examined in our study. Interfaces in bilayer solid electrolytes may serve
as weak interfaces (Fig. 4g and 4i) and could be engineered to act as dendrite-redirection path.

**Modification to the manuscript:**

Page 18: However, the presence of transverse voids can reduce the ionic conductivity of the solid
electrolyte. Therefore, in battery design, there is a trade-off between introducing transverse voids
to deflect dendrite propagation and the resulting decrease in ionic conductivity. According to the
observations of Ning et al.⁵, Swamy et al.⁷ and Xue et al.⁸, most dendrites tend to initiate and
propagate along the edges of the solid electrolyte due to current concentration. Hence, we believe
that engineering transverse voids near the edge of the solid electrolyte could provide a reasonable
balance between maintaining a relatively high ion flux and ensuring safety. Interfaces in bilayer
solid electrolytes may also serve as natural voids (Fig. 4g and 4i) and could be engineered to act
as crack-redirection path while maintaining chemical stability with electrodes.⁹⁻¹¹

Referee #2 (Remarks to the Author):

This work reports the fracture process in garnet electrolytes driven by lithium dendrites using
multiscale cryo S/TEM and modelling, to understand the long-standing debate on “soft-
penetrates-hard” failure. They revealed that hydrostatic stress buildup within the confined
lithium is the main reason for lithium penetration in solid electrolyte, rather than electron
leakage or isolated lithium nucleation ahead of the tip. The authors proposed a strategy using
geometrically engineered elliptical voids within the electrolyte to redirect dendrite growth
paths and prevent short-circuiting. The results are interesting. However, this reviewer feels that
the data presented are not convincingly enough to justify their correctness at this stage. And,
the chemical environment or surface free energy of the solid electrolyte differs from its bulk
properties, which may induce distinct dendrite growth mechanisms at the surface compared to
the interior. The authors' in-plane battery configuration cannot accurately represent real
conditions. Therefore, this manuscript is not recommended for publication in Nature. Additional
comments are as follows:

**Authors' response:** We sincerely thank the reviewer for his/her compliments and constructive
comments.

To address the relevance of the in-plane experiments here to real battery configurations, we have
now performed a full set of new experiments using a Li/LLZTO/Li symmetric cell configuration.
Specifically, we repeated all cryogenic characterizations on symmetric cells, including cryogenic
EBSD characterization of fractured LLZTO (Fig. S5-S6), three-dimensional fracture reconstruction
(Fig. S9-S10), TKD characterization of the Li dendrite (Fig. S17) and 3D micromechanical fracture
modelling (Fig. S36). These new measurements reproduce the key mechanistic features observed
in the in-plane cell, demonstrating that our conclusions are not specific to the geometry but are
intrinsic and general features of lithium-driven fracture in garnet LLZTO solid electrolyte. The
complete symmetric-cell dataset is provided in the Supplementary Information.

We also quantified the residual stresses induced by the Vickers indent and assessed their
potential influence on crack deflection. To do this, we compared the Raman peak shift (Fig. S32)
near the indent with that in the undeformed region and employed a multibeam optical stress
sensor (Fig. S33) to quantify the residual stress around the Vickers indent. All measurements
consistently show a small residual stress of approximately 1 MPa, which our fracture modelling
(Fig. S33) indicates that this level of residual stress has a negligible influence on the observed
crack deflection.

Regarding potential artefacts introduced by focused ion beam (FIB) milling, we separately
assessed possible damage to LLZTO and to lithium metal. To evaluate electron-beam-induced

degradation of LLZTO, we provide a so far unpublished cryogenic atom probe tomography results
from cryo-FIB-prepared specimens, which show atomic compositions identical to those
determined by X-ray diffraction. To examine whether FIB milling could induce strain
rearrangement, we show that the 1- μm -thick lift-out lamella used for lattice-rotation
measurements maintains its mechanical integrity from ion milling. Moreover, the thickness of the
region directly exposed to the ion-milling beam is substantially smaller than that of the intact
region.

We believe by performing additional experiments, simulations and analyses, we have fully
addressed all concerns raised by the reviewer, and justified the correctness of our results and
methods. Detailed responses to the reviewer's specific comments are provided below.

Referee 2 comment 1: The authors used phase-field simulations to predict 600 MPa hydrostatic
stress in Li, but no direct experimental measurement is provided.

**Authors' response:**

We thank the reviewer for raising this point. In the main manuscript (page 12), we used phase-
field fracture modeling to examine that the stress state within the Li dendrite and to establish
that dendrite propagation is governed predominantly by hydrostatic stress rather than by
deviatoric stress. These simulations also show that Li dendrite growth behavior is largely
insensitive to the yield stress of Li metal.

In discussing these simulations, we reported that the peak hydrostatic stress inside the Li dendrite
reaches approximately 600 MPa immediately before crack propagation, after which it decreases
to around 435 MPa during steady dendrite growth. We note that direct experimental
quantification of the stress state inside a nanoscale Li dendrite is extremely challenging, and to
our knowledge there are no published measurements of stresses within a Li dendrite. However,
Athanasiou *et al.*⁴ reported experimental measurements of stresses ahead of propagating
dendrites within the solid electrolyte, and we compared our simulated (Fig. S18) principal stress
difference ($\sigma_1 - \sigma_2$) with their measurements. The simulation results show good agreement with
an experimental results reported by Athanasiou *et al.*⁴ This consistency supports the physical
soundness and loading representation of our model.

[REDACTED]

**Figures from Athanasiou *et al.*⁴** showing the difference in principal stresses at the tip of a progressing
dendrite. The color bars correspond to the stress levels calculated from the measured retardance maps.

**Modification to the manuscript:**

Page 13: The simulated stress distribution ahead of the propagating lithium dendrite matches well
with the experimental measurements reported by Athanasiou-Fincher *et al.*⁴

1
 2 Figure S18. Distribution of the difference between the principal stresses $\sigma_1 - \sigma_2$ during lithium dendrite
 3 propagation. The simulation results show good agreement with the experimental measurement reported by
 4 Athanasiou-Fincher *et al.*⁴

Referee 2 comment 2: Page 3, "...provide indirect, greyscale-based imaging of lithium and cracks".
This is an unfair claim. OM, X-ray, SEM are direct observations as the work presented here. The
difference is resolution and analysis ability.

**Authors' response:**

We thank reviewer for this suggestion. We have rephrased this sentence in the main manuscript
to increase clarity.

**Modification to the manuscript:**

Page 3: Efforts to distinguish these failure mechanisms have primarily relied on optical microscopy,
X-ray tomography, and scanning electron microscopy (SEM) to observe lithium penetration.¹²⁻¹⁶
However, these techniques have limited ability to resolve the lithium/crack morphology, or to
resolve the microstructure of lithium at the dendrite tip due to resolution limit and analysis ability.

Referee 2 comment 3: The in-plane cell geometry applied here could not represent the other
geometries. How the observations here are representative is questionable. Moreover, a 2D slab
solid electrolyte model is demonstrated here, however, the 3D lithium growth in real electrolytes
could be different.

**Authors' response:**

We clarify here that the dendrite is growing in the plane of the electrolyte as shown in Fig. S11,
rather than on the surface. In response to a separate reviewer comment (*Referee 1, Comment 1*),
we fully comply with this concern and we have therefore now performed a full additional suite of
symmetrical cell experiments, including cryogenic EBSD characterization of fractured LLZTO (Fig.
S5-S6), 3D fracture reconstruction (Fig. S9-S10), TKD characterization of the Li dendrite (Fig. S17)
and 3D micromechanical fracture modelling (Fig. S36).

- • Fig. S5-S6 show the fracture pattern of the solid electrolyte induced by the growth of Li
dendrite in the symmetrical cell and allow direct comparison with the in-plane cell
results in Fig. 1. Fig. S6d-S6f show the crystallographic lattice orientation around the
fractured solid electrolyte. Consistent with the in-plane configuration, we again observe
both intergranular and transgranular fracture modes of the solid electrolyte induced by
Li dendrite propagation in the symmetrical cell, with approximately 20-30% of the
fracture events being transgranular. However, it is not possible to accurately quantify the
intergranular-transgranular ratio or determine the crack-deflection angle because the
fracture contour is not well-defined in the symmetric-cell geometry, in contrast to the
clearly resolved crack paths shown in Fig. 1e.
- • Fig. S9-S10 present the 3D fractography of the solid electrolyte in the symmetrical cell and
enable direct comparison with the in-plane cell results in Fig. 2. The reconstructed Li
dendrite morphology in Fig. S10 exhibits a tortuous geometry similar to that shown in Fig.
2b in the main manuscript. Notably, the seemingly isolated Li dendrite #1-#3 observed in
Fig. S10e (slice 350) were revealed through the full 3D reconstruction to originate from a
continuous dendrite that is already visible in slice 1 (Fig. S10e). Without 3D reconstruction
at cryogenic temperature, such an observation could be misinterpreted as evidence for
isolated Li nucleation and propagation within the LLZTO electrolyte.
- • Fig. S17 demonstrates the microstructure of Li dendrites in the symmetric cell and allows
direct comparison with the in-plane cell results in Fig. 3. Figs. S17c and S17d present the
lattice orientation map and grain reference orientation deviation (GROD) angle map of the
Li dendrite, respectively. The Li dendrite grain size is of the same order of magnitude as
that observed in Fig. 3. The GROD map in Fig. S17d reveals minimal lattice rotation across
most of the dendrite volume. Modest orientation gradients are observed only near the
lithium/LLZTO interface, whereas the dendrite interior exhibits negligible gradients,
indicating the absence of significant plastic strain. All these observations are consistent

with those in Fig. 3 and further support our conclusion that hydrostatic stress, rather than
plastic deformation, serves as the dominant driving force responsible for the “soft-
penetrates-hard” behavior during lithium dendrite growth.

- • Fig. S36 shows the 3D micromechanical fracture modeling results for the symmetric cell
configuration and allows direct comparison with the 2D modelling results in Fig. 3. The
hydrostatic stress buildup within the Li dendrite remains orders of magnitude higher than
the von Mises stress, consistent with the 2D simulation results. These 3D modelling results
therefore reinforce our conclusion that the hydrostatic stress contribution within the Li
dendrite dominates the mechanical driving force for the dendrite-induced fracture of the
LLZTO electrolyte.

All these results from Figs. S5-S6, S9-S10, S17 and S36 together demonstrate that the Li dendrite
formation behavior observed in the in-plane cell is consistent with that occurring in the
symmetrical cell.

**Modification to the manuscript:**

We note that this comment requires the same revision as addressed in *Referee 1, Comment 1*. To
avoid redundancy, the modified figures (S5–S6, S9–S10, S17, and S36) are included under *Referee*
*1, Comment 1* or in the Supporting Information.

**Page 7:** The lithium-plating-induced fracture behavior of the Li/LLZTO/Li symmetrical cell was
also examined, as shown in Figs. S5 and S6. Both intergranular and transgranular fractures features
were observed, with approximately 20-30% of the fracture events being transgranular. However,
it is not possible to accurately quantify the intergranular-transgranular ratio or determine the crack-
deflection angle because the fracture contour is not well-defined in the symmetric-cell geometry,
in contrast to the clearly resolved crack paths shown in Fig. 1e.

**Page 9:** The three-dimensional fractography of LLZTO in the Li/LLZTO/Li symmetric cell was
further analyzed, as presented in Figs. S9 and S10. The reconstructed lithium dendrite morphology
in Fig. S10 exhibits a tortuous geometry similar to that shown in Fig. 2b. Notably, the seemingly
isolated lithium dendrite #1 to #3 (Fig. S10e, slice 350) originate from a single continuous dendrite
(Fig. S10e, slice 1). Without three-dimensional reconstruction at cryogenic temperature, such an
observation could be misinterpreted as evidence for isolated lithium nucleation and propagation
within the LLZTO electrolyte.

**Page 13:** The microstructure of the lithium dendrite growth within the solid electrolyte under the
symmetric-cell configuration is shown in Fig. S17. The GROD map also exhibits minimal lattice
rotation, comparable to that observed in Fig. 3c.

**Page 25:** These three-dimensional simulations (Fig. S36) verify the two-dimensional results and
confirm that the predicted fracture behavior remains consistent across both geometries.

**Modification to the Method section:**

Page 20: Electrochemical Impedance Spectroscopy (EIS) data were recorded in the frequency
range between 10 Hz and 7 MHz with an amplitude of 50 mV using a SP200 impedance analyzer
(Bio-Logic).

Page 20: Lithium dendrite growth through the symmetric cell configuration was cycled using the
same potentiostat equipped with a pressure stand (Imada Inc.). Before applying the bias, the
symmetric cell was heated to 130 °C using a heating sleeve (RS Components Ltd.) to improve the
interfacial contact between the lithium metal and the solid electrolyte. After short-circuiting, the
lithium metal was removed using sandpaper with a grit size of 1200. The short-circuited solid
electrolyte was then taken out of the glovebox, soaked in epoxy overnight for curing, and
subsequently polished to the region where features resembling lithium dendrites could be observed,
as shown in Figs. S5e and S5f. Fig. S5d schematically illustrates the sample preparation procedure.
Fig. S5e shows the surface after rough polishing with 320-grit sandpaper, while Fig. S5f shows the
result after fine polishing using a 0.1 μm SiO₂ polishing suspension. The lithium metal on the
plating side, where dendrite growth occurred, could be easily peeled off by hand in the results
shown in Figs. S9, S10 and S17. Therefore, no sandpaper was used to remove the lithium electrode,
in contrast to the procedure used for the samples shown in Figs. S5–S6.

Referee 2 comment 4: The confirmation of the presence of lithium in the crack was done by STEM-
EELS of the FIB lamella samples. As already been known that lithium metal could be reduced by
ion beam or electron beam bombardment. Therefore, complicated reactions could occur during
the FIB processing. This may not be a suitable way to reveal the original status of the matter inside
the crack. Direct detection of lithium in the thicker SEM samples is more reliable. Moreover, raw
data of the EELS spectra is suggested. MSA analysis may bring artifacts and details of the data
processing was not provided.

**Authors' response:**

As the reviewer correctly noted, Li metal can indeed be readily reduced from Li ions in the solid
electrolyte when exposed to electron-beam irradiation at room temperature—a behavior that
has been reported in multiple studies.¹⁶⁻¹⁸ However, all electron microscopy characterizations
presented in the main manuscript and supporting information were conducted under cryogenic
conditions (−190 °C for cryo-FIB and −150 °C for cryo-TEM). Across all these characterizations, no
detectable formation of metallic Li due to beam–sample interactions were observed.

To address the reviewer’s concern about the “complicated reactions could occur during the FIB
processing”, we present a so far unpublished result here (*Figure for Referee 2 comment 4*) which
we are able to perform atomic probe tomography on the $\text{Li}_{6.6}\text{La}_3\text{Zr}_{1.6}\text{Ta}_{0.4}\text{O}_{12}$ solid electrolyte. The
atomic composition ratio of Li/La was confirmed to be 2.2 by the manufacturer (Toshiba
Manufacturing Co., Ltd.). The sample was prepared entirely using a focused ion beam under
cryogenic conditions. The atomic ratio of Li/La measured by mass spectrometry was also 2.2,
consistent with the manufacturer’s specification. More importantly, we didn’t find any Li
enrichment in the prepared APT needle which demonstrate the FIB preparation under cryogenic
condition won’t lead to any detectable reduced Li metal from the solid electrolyte.

[REDACTED]

To further address the reviewers' concern regarding the possibility of "complicated reactions
occurring during FIB processing," we prepared an additional TEM lamella located approximately
5 μm ahead of the dendrite front, as shown in Fig. S14. This lamella reveals several triple junctions
that are commonly observed in hot-pressed ceramic pellets. EELS mapping at cryogenic
temperature confirms that these triple junctions are unfilled (Figs. S14b and S14c). If complicated
reactions had occurred during cryogenic FIB processing, as concerned by the reviewers, Li metal
filling should have been detectable within these triple junctions—sites that represent the most
energetically favorable locations for Li-ion reduction. However, we didn't find Li metal in the triple
junction. Therefore, together with the APT results, our findings indicate that FIB preparation at
cryogenic temperature does not induce any detectable reduction of Li metal from the solid
electrolyte.

Regarding to the comment "Direct detection of lithium in the thicker SEM samples is more reliable".
First, with an SEM image in Fig. S11, we have already shown the Li filling condition (in the original
manuscript) at 5 μm ahead of crack tip and at the exact dendrite tip through cross sectional image
in thick SEM samples. Second, the results in Fig. S17 and Fig. 3 are transmission Kikuchi diffraction
results record in scanning electron microscope in thicker SEM samples (1 μm thick to enable
sufficient elastic scattering) as suggested by the reviewer.

We agree with the reviewer's comments regarding the multivariate statistical analysis (MSA). To
eliminate any possibility of processing artifacts, all EELS spectra are now reported in their raw
form after a standard power-law background subtraction. In addition, we included in the
Supporting Information further details on the use of MSA to quantify the Li counts, demonstrating
its capability to distinguish Li dendrite signals from those of the LLZTO region.

**Modification to the manuscript:**

Page 20: The detrimental interaction between the electron beam and the solid electrolyte is strongly
suppressed at cryogenic temperatures. No electron-beam-induced lithium nucleation was observed
under cryogenic conditions, in contrast to the artifacts frequently encountered at room
temperature.^{16,18} Moreover, the elemental distribution within the solid electrolyte was successfully
mapped by atom probe tomography, using samples prepared following the same cryogenic
preparation procedure (work under preparation).

Page 21: In order to facilitate comparison with EELS spectra reported in literatures, we opt to
display raw EELS spectra from selected areas in Fig S12b, S14d and S14e.

Page 21: For lithium count maps shown in Fig. 2f and 2g, power law background was modelled
for components 1 and 2, with respective fitting windows of (45, 50) eV and (45, 57) eV. The
integration window was kept to (57, 67) eV. As evidenced in Fig. S12, the Li-K edge onsets of the
LLZTO and the Li/LiOH phases are different. The quantification of lithium is hence facilitated by
multivariate statistical analysis¹⁹, where most of the spatial variance in EELS signal can be
expressed in the component 1 and component 2. As shown in Fig. S12c-f, component 1 is primarily
located in the dendrite area, and the spectral feature is LiOH-like; component 2 relates to the

LLZTO area surrounding the dendrite, with LLZTO-like spectral feature. The component 3 no
longer have spectral features typical of a physical spectrum, as they represent minor differential
signals to modify the two leading components. This observation confirms the dominance of the
Li/LiOH and LLZTO phase in this area.

Figure S12. (a) EELS Li count map at the lithium dendrite tip (same dataset as in Fig. 2g). (b) Integrated
EELS spectra from LLZTO and the lithium dendrite obtained from the raw EELS dataset. The spectrum from
the lithium dendrite exhibits characteristic features of LiOH, indicating contamination introduced during
sample transfer from cryo-FIB to cryo-TEM. TKD-SEM measurement results performed prior to transfer
confirm that the lithium dendrite consisted of pristine metallic lithium, as supported by the high
confidence index in the EBSD analysis in Fig. 3B. Accordingly, throughout the manuscript, LiOH-related
EELS peaks are attributed to contamination of originally pristine lithium. (c-f) Multivariate statistical
analysis applied to the dataset in Fig. S12a, showing the three leading spectral components (c) and their
corresponding weighting maps (d-f). Further methodological details are provided in the Methods section
(Cryogenic S/TEM). Data source: in-plane cell.

Referee 2 comment 5: Again, the conclusion related to lattice distortion or rotation relies on the
analysis of the FIB samples. Ion milling may cause the release or rearrangement of the strain. How
the results are representative for the real situation is doubtful.

**Authors' response:**

To address this concern, Fig. S35 presents the lift-out lamella used for the TKD measurement
shown in Fig. 3b. The lamella thickness was kept at approximately 1 μm to ensure sufficient
mechanical rigidity and to avoid any bending-induced strain. Moreover, at this thickness, the Ga-
ion-milled region is sufficiently thin compared with the part of the lamella that remained intact
and unaffected by the Ga^+ focused ion beam.

Moreover, several works from Prof. Jürgen Janek's team²⁰⁻²² shown the lattice orientation of
plated Li and Na metal by milling the sample through FIB under cryogenic temperature. One work
from Fuchs *et al.*²¹ reported that "no changes in grain orientation and grain size are observed
after a second milling step, demonstrating that the FIB preparation does not affect the
microstructure of sodium". The results in Fig. S7 in Fuchs *et al.*²¹ support this statement.

To further address this concern, we present a 3D EBSD reconstruction obtained from sequential
FIB slicing performed at room temperature. These data were reproduced (with permission from
the authors) from a master's thesis conducted under the supervision of Dr. Stefan Zaefferer, who
is also a co-author of this manuscript. *Figure for Referee 2 comment 5* shows the lattice
orientation and kernel average misorientation (KAM) evolution of a 30% cold-rolled Mg-3 wt% Y
alloy through the thickness direction, reconstructed from 35 FIB slices with a thickness of 400 nm
each. The presence of shear bands and residual-stress concentrations in the KAM maps
demonstrates that FIB cutting did not cause any measurable stress release or strain
rearrangement. Moreover, these Mg-3 wt% Y samples (melting point = 923 K) were FIB-milled at
room temperature (298 K). In contrast, the Li dendrite (melting point = 453 K) was prepared at a
cryogenic temperature of 83 K. Given that the Mg alloy exhibited no measurable lattice rotation
or strain rearrangement when FIB-cut and EBSD-scanned at 32% of its melting point, we have
strong grounds to claim that Li would likewise not undergo any measurable strain rearrangement
when FIB preparation and EBSD measurements are performed at only 18% of its melting point.

As a summary, from both our results and other literature, we see no evidence that ion milling
through focused ion beam induces measurable lattice rearrangement, especially at cryogenic
temperature.

**Figure for Referee 2 comment 5.** 3D auto-slice view and EBSD analysis of a 30% cold-rolled Mg-3 wt% Y
 alloy. (a) Lattice orientation map. (b) Kernel average misorientation (KAM) map. (c-f) Selected slices from
 the 3D voxel reconstruction showing the continuity of deformation features prepared by FIB milling, such
 as lattice orientation gradients and shear bands.

**Modification to the manuscript:**

Page 21: Fig. S35 shows the TKD lamella, which maintains its mechanical integrity without any
observable bending or distortion induced by ion-milling preparation. Moreover, because the
sample was prepared using Ga⁺ FIB at cryogenic temperature, strain rearrangement during ion
milling is expected to be strongly suppressed and therefore experimentally negligible, as reported
in several previous studies.²⁰⁻²²

Figure S35. Ga⁺ FIB lift-out lamella prepared for transmission Kikuchi diffraction (TKD) analysis. The
lamella was maintained at a thickness of approximately 1 μm to ensure sufficient electron scattering and to
preserve its mechanical integrity, thereby preventing bending or distortion-induced strain during ion-milling
preparation.

Referee 2 comment 6: The authors employed optical microscopy to study the top-view growth
direction of lithium dendrites near indent arrays, but this method has limitations. Since lithium
dendrites can propagate in 3D within the solid electrolyte, a 2D top-view observation cannot
confirm whether the dendrites actually intersect with the indents in the depth direction.
Additionally, the resolution of optical microscopy is too low to definitively determine whether the
observed features are indeed lithium dendrites or cracks, as claimed by the authors.

**Authors' response:**

We thank the reviewer for this valuable comment. We have included a cryogenic SEM image (Fig.
S30) to illustrate the interaction between the Li dendrite and the Vickers indent. The Li dendrite
is observed to propagate toward the crack (emanating from the Vickers indent) and subsequently
deflect by approximately 45°. Fig. S30b shows the corresponding cross-sectional fracture image
of the solid electrolyte induced by Li dendrite growth. The lamella was intentionally thinned to a
thickness of approximately 100 nm. If the crack had not been fully filled, the background behind
the lamella would have been visible through the unfilled region. A high-resolution image of a
Vickers indent has now been included in Fig. 4c and 4d.

**Modification to the manuscript:**

Page 16: The post-mortem cryogenic SEM image (Fig. 4d) shows that the lithium dendrite
propagates toward the cracks emanating from the Vickers indent and subsequently deflects by
approximately 45°, following the pre-existing crack induced by Vickers indent. The subsurface
interaction between the lithium dendrite and the LLZTO solid electrolyte is shown in Fig. S29.

Figure S29. (a) Deflection of a lithium dendrite after interacting with Vickers indents. (b) Cross-sectional
fractography of LLZTO induced by lithium plating in the region highlighted by the red square in (a). The
region of interest was intentionally thinned down to ~ 100 nm to demonstrate that it is fully filled; otherwise,
the background would be visible through the dry crack. Data source: in-plane cell.

**Fig. 4. Tailor lithium dendrite growth through engineered voids.** (a, b) Top-view optical microscope
image showing lithium dendrite growth (a) before and (b) after encountering arrays of Vickers indents. The
dendrite deflects upon reaching cracks emitting from the Vickers indent, altering its trajectory. (c) Zoom-in
optical microscope image of a representative Vickers indent. (d) Zoom-in SEM image showing the
interaction between the crack network and the lithium dendrite. (e-h) Phase-field simulation of lithium
dendrite interaction with (e) circular and (g) transverse voids embedded in the solid electrolyte, showing
hydrostatic stress in lithium and maximum tensile stress in LLZTO during dendrite propagation. Transverse
voids promote deflection, while circular voids allow straight propagation. (i) Schematic of mechanically
driven fracture during lithium dendrite propagation, leading to failure in solid-state batteries. (j) Schematic
of the stress field in the lithium dendrite and solid electrolyte, with hydrostatic pressure buildup in lithium
dendrite and tensile stress in the surrounding solid electrolyte. CAM: cathode active material; SE: solid
electrolyte.

Referee 2 comment 7: The authors artificially created voids using Vickers indents and observed
that the growth direction of lithium dendrites deflected upon reaching the indent array. However,
their conclusion seems overly speculative. An indent is not simply a void—it primarily consists of
the indentation core and the surrounding deformation zone. The indentation core is the region
directly formed by the indenter's action, with its shape and dimensions closely related to the
indenter's geometry and loading conditions. The surrounding deformation zone arises from
plastic flow and stress transfer in the material during the indentation process. Therefore, the
possibility cannot be ruled out that the deflection of lithium dendrites is caused by their
interaction with the deformation zone around the indent rather than by the voids alone.

**Authors' response:**

We intended the cracks generated by the indents to be considered as transverse voids, rather
than the residual indents themselves. We thank the reviewer for bringing this point up. High-
resolution images of the interaction between Li dendrite and sharp cracks induced by Vickers
indent are added as Fig. 4. Related to the deformation and residual stress fields that surround the
indents, we agree with the reviewer that the stress field can also influence dendrite propagation,
as reported by Fincher *et al.*³, who showed that stresses on the order of hundreds of MPa can
lead to Li-dendrite deflection in garnet solid electrolytes. However, in our experiments we
observed cracks around the Vickers indents, which are expected to relax the residual stress. This
interpretation is supported by our residual stress measurements, which show less than 1 MPa of
stress remaining around the indent. In response to a separate reviewer comment (*Referee 1,*
*Comment 4*), our detailed explanation is provided below:

- • To estimate the residual stress induced by plastic carriers, we performed cryo-EBSD
mapping around the Vickers indent. If plastic flow had occurred around the indent,
variations in the kernel average misorientation (KAM) map (proportional to geometrical
necessary dislocation density) should have been observed. However, the results in Fig.
S31b-c show no discernible trend or increase in KAM values. Furthermore, the sample
surface was mechanically polished (down to a 0.1 μm SiO_2 suspension) to remove
approximately 20 μm of the surface layer and to investigate potential subsurface residual
stress induced by plastic flow. The subsurface results (Fig. S31e-f) were consistent with
those from the surface region, showing no measurable variation in KAM values around
the Vickers indent.
 - • To estimate the residual stress induced by elastic response, we first compared the Raman
spectra obtained near the Vickers indent—where residual stress may exist—with those
from the undeformed region of the garnet solid electrolyte. As shown in Fig. S32, no

measurable peak shift was observed between the region close to the Vickers indent and
the undeformed region. Second, we employed a more sensitive technique—multibeam
optical stress sensor—which can detect a change in radius of curvature on the order of 20
4 km, corresponding to a strain variation of approximately 0.01% in a 0.5 mm-thick LLZTO
solid electrolyte. Fig. S33a shown the schematic of experimental setup and Fig. S33b show
the curvature change before and after inducing a 7*4 array Vickers indent (spacing 150
7 μm) into the solid electrolyte. Plugging the curvature change into Stoney equation, we get
a residual compression around 1MPa. By incorporating this value into our phase field
modelling (Fig. S33d and S33e), we found the measured residual stress won't influence
the crack propagation path in our work.

As a summary, deflection of lithium dendrites is indeed cause by the interaction with the cracks
induced by the Vickers indent. The deformation zone plays a negligible role on the interaction of
Li dendrite with the cracks.

**Modification to the manuscript:**

We note that this comment requires the same revision as addressed in *Referee 1, Comment 4*. To
avoid redundancy, the modified figures (S31-S33) are included under *Referee 1, Comment 4* or in
the Supporting Information.

Page 16: We note that the residual stress in polycrystalline LLZTO is substantially relaxed by
cracking, and our measurements indicate that only ~ 1 MPa remains. Previous studies³ have shown
that dendrite deflection in garnet electrolyte requires stresses of the order of 100 MPa, whereas the
residual stress surrounding our Vickers indents is two orders of magnitude lower. Accordingly, our
fracture modelling indicates that this level of stress has a negligible influence on the observed
crack deflection. Detailed measurements and calculations are provided in Figs. S31-S33.

Referee 2 comment 8: The authors suggest that artificially introduced voids can alter dendrite
growth direction and prevent short circuits, but such voids may simultaneously reduce the ionic
conductivity of the solid electrolyte. Additionally, the authors propose that "interfaces in
multilayer solid electrolytes may serve as natural voids". What is the underlying rationale for this
claim?

**Authors' response:**

We thank the reviewer for this comment. Indeed, the reviewer is correct that the presence of
shape-edged voids can reduce the ionic conductivity of the solid electrolyte. Therefore, in battery
design, there is a trade-off between introducing transverse voids to deflect dendrite growth and
the resulting decrease in ionic conductivity. According to the observations of Ning *et al.*²³, Swamy
*et al.*⁷ and Xue *et al.*⁸, most dendrites tend to initiate and propagate along the edges of the solid
electrolyte due to current concentration. Hence, we believe that engineering voids near the edge
of the solid electrolyte could provide a reasonable balance between maintaining a relatively high
ion flux and ensuring safety.

Regarding the second comment, the bilayer solid electrolyte strategy has recently been employed
in the community to mitigate electrochemical instability arising from the direct contact between
the solid electrolyte and the cathode or anode.⁹ In addition, mechanically compressing two
different materials without annealing typically leads to the formation of a weak interface
containing voids. The distribution, density, and curvature of these voids can be engineered.¹⁰
Therefore, we initially proposed that the voids formed at the bilayer solid electrolyte interface
could serve as crack-deflection paths while maintaining chemical stability with the electrodes. We
have rephrased this sentence to increase clarity.

**Modification to the manuscript:**

Page 18: However, the presence of transverse voids can reduce the ionic conductivity of the solid
electrolyte. Therefore, in battery design, there is a trade-off between introducing transverse voids
to deflect dendrite propagation and the resulting decrease in ionic conductivity. According to the
observations of Ning *et al.*⁵, Swamy *et al.*⁷ and Xue *et al.*⁸, most dendrites tend to initiate and
propagate along the edges of the solid electrolyte due to current concentration. Hence, we believe
that engineering transverse voids near the edge of the solid electrolyte could provide a reasonable
balance between maintaining a relatively high ion flux and ensuring safety. Interfaces in bilayer
solid electrolytes may also serve as natural voids (Fig. 4g and 4i) and could be engineered to act
as crack-redirection path while maintaining chemical stability with electrodes.⁹⁻¹¹

Referee 2 comment 9: While I appreciate the authors' efforts to investigate the origin of dendrite
penetration, the findings remain largely phenomenological. The observed results may represent
either strain-relaxed states or artifacts induced by ion beam modification, rather than capturing
the material's original state. Consequently, the claim of uncovering the 'origin' appears
overstated. Overall, this work offers only incremental advancement compared to prior studies in
the field.

**Authors' response:**

We thank the reviewer for the valuable comments and constructive critique. However, we
respectfully disagree with the reviewer's statement "only incremental advancement". We have
carefully addressed all concerns by incorporating extensive new experiments, including all the
cryogenic characterization (3D reconstruction, EBSD, TKD) in symmetrical cell geometry and
complementary micromechanical modelling. These additional results consistently support our
original conclusions and, we hope, convincingly demonstrate that the reported mechanistic
insights are intrinsic to lithium-plating-driven fracture.

Reference:

- (1) Puls, S.; Nazmutdinova, E.; Kalyk, F.; Woolley, H. M.; Thomsen, J. F.; Cheng, Z.; Fauchier-
Magnan, A.; Gautam, A.; Gockeln, M.; Ham, S.-Y.; et al. Benchmarking the reproducibility of all-
solid-state battery cell performance. *Nature Energy* **2024**, *9* (10), 1310-1320. DOI:
10.1038/s41560-024-01634-3.
- (2) Liu, X.; Garcia-Mendez, R.; Lupini, A. R.; Cheng, Y.; Hood, Z. D.; Han, F.; Sharafi, A.; Idrobo, J.
C.; Dudney, N. J.; Wang, C.; et al. Local electronic structure variation resulting in Li 'filament'
formation within solid electrolytes. *Nature Materials* **2021**, *20* (11), 1485-1490. DOI:
<https://doi.org/10.1038/s41563-021-01019-x>.
- (3) Fincher, C. D.; Athanasiou, C. E.; Gilgenbach, C.; Wang, M.; Sheldon, B. W.; Carter, W. C.;
Chiang, Y. M. Controlling dendrite propagation in solid-state batteries with engineered stress.
*Joule* **2022**, *6* (12), 2794-2809. DOI: <https://doi.org/10.1016/j.joule.2022.10.011>.
- (4) Athanasiou, C. E.; Fincher, C. D.; Gilgenbach, C.; Gao, H.; Carter, W. C.; Chiang, Y.-M.; Sheldon,
B. W. Operando measurements of dendrite-induced stresses in ceramic electrolytes using
photoelasticity. *Matter* **2024**, *7* (1), 95-106. DOI: 10.1016/j.matt.2023.10.014 (accessed
2025/03/24).
- (5) Ning, Z.; Li, G.; Melvin, D. L. R.; Chen, Y.; Bu, J.; Spencer-Jolly, D.; Liu, J.; Hu, B.; Gao, X.;
Perera, J.; et al. Dendrite initiation and propagation in lithium metal solid-state batteries. *Nature*
**2023**, *618* (7964), 287-293. DOI: 10.1038/s41586-023-05970-4.
- (6) Chen, C.; Wang, Z.; Suo, Z. Flaw sensitivity of highly stretchable materials. *Extreme*
*Mechanics Letters* **2017**, *10*, 50-57. DOI: <https://doi.org/10.1016/j.eml.2016.10.002>.
- (7) Swamy, T.; Park, R.; Sheldon, B. W.; Rettenwander, D.; Porz, L.; Berendts, S.; Uecker, R.;
Carter, W. C.; Chiang, Y.-M. Lithium Metal Penetration Induced by Electrodeposition through
Solid Electrolytes: Example in Single-Crystal $\text{Li}_6\text{La}_3\text{ZrTaO}_{12}$ Garnet. *Journal of The Electrochemical*
*Society* **2018**, *165* (16), A3648. DOI: 10.1149/2.1391814jes.
- (8) Xue, D.; Fincher, C.; Fang, R.; Sheldon, B. W.; Chen, L.-Q.; Zhang, S. Dynamic interplay of
dendrite growth and cracking in lithium metal solid-state batteries. *Journal of the Mechanics*
*and Physics of Solids* **2025**, *202*, 106197. DOI: <https://doi.org/10.1016/j.jmps.2025.106197>.
- (9) Ye, L.; Li, X. A dynamic stability design strategy for lithium metal solid state batteries. *Nature*
**2021**, *593* (7858), 218-222. DOI: 10.1038/s41586-021-03486-3.
- (10) Hu, B.; Zhang, S.; Ning, Z.; Spencer-Jolly, D.; Melvin, D. L. R.; Gao, X.; Perera, J.; Pu, S. D.;
Rees, G. J.; Wang, L.; et al. Deflecting lithium dendritic cracks in multi-layered solid electrolytes.
*Joule* **2024**, *8* (9), 2623-2638. DOI: 10.1016/j.joule.2024.06.024.
- (11) Yu, Z.; Gan, C.; Mijailovic, A. S.; Stone, A.; Hurt, R.; Pernia, C. L.; Xiao, X.; Shi, C.; Sheldon, B.
35 W. Lithium Dendrite Deflection at Mixed Ionic–Electronic Conducting Interlayers in Solid
Electrolytes. *Advanced Energy Materials* **2025**, *15* (13), 2403179. DOI:
<https://doi.org/10.1002/aenm.202403179>.
- (12) Kazyak, E.; Garcia-Mendez, R.; LePage, W. S.; Sharafi, A.; Davis, A. L.; Sanchez, A. J.; Chen, K.-
H.; Haslam, C.; Sakamoto, J.; Dasgupta, N. P. Li Penetration in Ceramic Solid Electrolytes:
Operando Microscopy Analysis of Morphology, Propagation, and Reversibility. *Matter* **2020**, *2*
(4), 1025-1048. DOI: <https://doi.org/10.1016/j.matt.2020.02.008>.
- (13) Hao, S.; Bailey, J. J.; Iacoviello, F.; Bu, J.; Grant, P. S.; Brett, D. J. L.; Shearing, P. R. 3D Imaging
of Lithium Protrusions in Solid-State Lithium Batteries using X-Ray Computed Tomography.

*Advanced Functional Materials* **2021**, *31* (10), 2007564. DOI:
<https://doi.org/10.1002/adfm.202007564>.
(14) McConohy, G.; Xu, X.; Cui, T.; Barks, E.; Wang, S.; Kaeli, E.; Melamed, C.; Gu, X. W.; Chueh,
4 W. C. Mechanical regulation of lithium intrusion probability in garnet solid electrolytes. *Nature*
*Energy* **2023**, *8* (3), 241-250. DOI: 10.1038/s41560-022-01186-4.
(15) Wang, T.; Chen, B.; Liu, Y.; Song, Z.; Wang, Z.; Chen, Y.; Yu, Q.; Wen, J.; Dai, Y.; Kang, Q.; et al.
Fatigue of Li metal anode in solid-state batteries. *Science* **2025**, *388* (6744), 311-316. DOI:
doi:10.1126/science.adq6807.
(16) Krauskopf, T.; Dippel, R.; Hartmann, H.; Peppler, K.; Mogwitz, B.; Richter, F. H.; Zeier, W. G.;
Janek, J. Lithium-Metal Growth Kinetics on LLZO Garnet-Type Solid Electrolytes. *Joule* **2019**, *3*
(8), 2030-2049. DOI: 10.1016/j.joule.2019.06.013.
(17) Zhu, C.; Fuchs, T.; Weber, S. A. L.; Richter, F. H.; Glasser, G.; Weber, F.; Butt, H.-J.; Janek, J.;
Berger, R. Understanding the evolution of lithium dendrites at $\text{Li}_{6.25}\text{Al}_{0.25}\text{La}_3\text{Zr}_2\text{O}_{12}$ grain
boundaries via operando microscopy techniques. *Nature Communications* **2023**, *14* (1), 1300.
DOI: 10.1038/s41467-023-36792-7.
(18) Peng, X.; Tu, Q.; Zhang, Y.; Jun, K.; Shen, F.; Ogunfunmi, T.; Sun, Y.; Tucker, M. C.; Ceder, G.;
Scott, M. C. Unraveling Li growth kinetics in solid electrolytes due to electron beam charging.
*Science Advances* *9* (17), eabq3285. DOI: <https://doi.org/10.1126/sciadv.abq3285>.
(19) Zhang, S.; Scheu, C. Evaluation of EELS spectrum imaging data by spectral components and
factors from multivariate analysis. *Microscopy* **2018**, *67* (suppl_1), i133-i141. DOI:
10.1093/jmicro/dfx091.
(20) Becker, J.; Fuchs, T.; Ortmann, T.; Kremer, S.; Richter, F. H.; Janek, J. Microstructure of
Lithium Metal Electrodeposited at the Steel| $\text{Li}_6\text{PS}_5\text{Cl}$ Interface in “Anode-Free” Solid-State
Batteries. *Advanced Energy Materials* **2025**, *15* (16), 2404975. DOI:
<https://doi.org/10.1002/aenm.202404975>.
(21) Fuchs, T.; Ortmann, T.; Becker, J.; Haslam, C. G.; Ziegler, M.; Singh, V. K.; Rohnke, M.;
Mogwitz, B.; Peppler, K.; Nazar, L. F.; et al. Imaging the microstructure of lithium and sodium
metal in anode-free solid-state batteries using electron backscatter diffraction. *Nature Materials*
**2024**, *23* (12), 1678-1685. DOI: 10.1038/s41563-024-02006-8.
(22) Haslam, C. G.; Fuchs, T.; Liao, D. W.; Becker, J.; Dasgupta, N. P.; Janek, J.; Sakamoto, J. The
Effect of Alloying Interlayers on Lithium Anode Morphology and Microstructure in “Anode-Free”
Solid-State Batteries. *ACS Energy Letters* **2025**, *10* (5), 2285-2291. DOI:
10.1021/acsenerylett.5c00149.
(23) Ning, Z.; Jolly, D. S.; Li, G.; De Meyere, R.; Pu, S. D.; Chen, Y.; Kasemchainan, J.; Ihli, J.; Gong,
C.; Liu, B.; et al. Visualizing plating-induced cracking in lithium-anode solid-electrolyte cells.
*Nature Materials* **2021**, *20* (8), 1121-1129. DOI: 10.1038/s41563-021-00967-8.

Referees' comments:

Referee #1 (Remarks to the Author):

Both reviewer comments had a number of similarities, including regarding how representative
the experimental geometry is and differences between the indents and true void
structure/stress state. The authors have carried out a robust and extensive series of additional
experiments to address the variety of comments. I appreciate the additional experiments in the
symmetric cell with different geometry, and in general the prior and new experimental results
are interesting.

The experimental findings in this paper provide new understanding of lithium growth causing
stress and fracture, and I think they are useful for the community. The experiments are difficult
and the results are high quality. Given the positive but somewhat reserved response from both
reviewers, I think the paper could be acceptable in Nature if the editors are sufficiently excited
about the paper.

**Authors' response:**

We thank the referee very much for the careful evaluation of our manuscript and for the
constructive and pertinent comments, which helped us profoundly to further improve our paper.
We appreciate the referee's recognition that the additional experiments have further
strengthened the work, and that the experimental findings provide new insight into lithium
growth-induced stress and fracture in garnet solid electrolytes. We also note the referee's
assessment that the experiments are technically challenging and that the resulting data are of
high quality making it acceptable for Nature. Below, we address the specific comments in detail.

Referee 1 comment 1: I do have one additional comment. My original comment on real solid-
state separators being thin was not addressed in accord with how it was originally intended. My
point was not necessarily that filaments would grow quickly through thin (20-30 micron) solid
state separators, but instead that with such thin separators there is much less volume over which
to carry out any sort of pore or void engineering strategy. In a 1-mm thick pellet, voids can be
engineered easily throughout this thickness. but in a 20-micron film, void creation might affect
the entire cross section of the film. I believe the authors should address this question in the text:
is void engineering realistic in 20 micron films that need to be produced over meters-square area
and must be highly uniform and ionically conductive? This is not addressed in the revision.

**Authors' response:**

We agree that implementing void engineering in the most ideal thin solid electrolytes (~20 μm)
presents practical complexities related to scalable fabrication. We think that one possible way to
implement void engineering is to use a bilayer solid state electrolyte system. We believe that, by
controlling particle size, stacking pressure from rolling, sintering temperature and time, interfaces
in bilayer solid electrolytes could be engineered as an intended mechanical weak spot to deflect
lithium dendrite growth. While our proof-of-concept demonstrated in this work was not
evaluated under a 20 μm thin solid electrolyte, we provide the fundamental design principle to
achieve safe operation. Future studies will therefore be required to assess the feasibility of void
engineering in solid-electrolyte films with thicknesses on the order of 20 μm while maintaining a
highly uniform and ionically conductive state. Yet, this further engineering implementation study
is beyond the scope of our current paper.

The revised manuscript explicitly presents the introduction of voids as a proof-of-concept,
intended to expand future design options for solid-state batteries. The discussion now
emphasizes the underlying design principle, namely, that introducing local mechanical defects
(such as voids, cracks, or weak interfaces) can influence and even guide lithium dendrite
propagation pathways. For example, interfaces in multilayer solid electrolytes may serve as
mechanically weak regions that promote lithium dendrite deflection while preserving ionic
transport and chemical stability at the lithium/solid-electrolyte interface. Insofar we fully comply
with the reviewer's suggestions and we have modified the text accordingly.

**Modification to the manuscript:**

Page 17: These results reveal that transverse voids could redirect the growth direction of lithium
dendrites by locally modifying the tensile stress distribution within the ceramic electrolyte. This
mechanistic understanding offers a proof-of-concept for implementing defect-based strategies to
control dendrite propagation and suppress short-circuiting in solid-state batteries.

Page 18: Mechanically guided redirection of dendrite propagation: As shown in Fig. 4a-4d,
transverse voids aligned perpendicular to the dendrite propagation direction redirect dendrite
growth paths and thus prevent short-circuiting. This proof of concept shows that introducing local
defects (e.g. voids, cracks, or weak interfaces) can effectively influence the dendrites' propagation
paths. To realize this concept in a thin solid electrolyte separators (ideally down to $\sim 20 \text{ nm}^1$),
interfaces in multilayer solid electrolytes could potentially be leveraged as mechanically weak
regions to redirect dendrite propagation.²⁻⁴ This approach provides a fundamental design principle
for mitigating dendrite induced short-circuiting while preserving overall ionic transport and
chemical stability. Scalable fabrication in such lower-dimensional engineering systems require
further investigated.

Referee #2 (Remarks to the Author):

Referee 2 comment 1: Both I and the other reviewer raised concerns regarding whether the
specially designed 2D in-plane geometry setup accurately represents real 3D scenarios. I
anticipate future readers sharing this concern. While the authors' revisions primarily addressed
this point, the core structure remains unchanged: the main text still focuses on 2D results,
relegating all 3D results to the SI. The minor revisions to the main text are insufficient to alleviate
potential reader doubts. At minimum, key 3D results should be included in the main figures.

**Authors' response:**

We thank the reviewer for this suggestion. We agree that demonstrating 3D symmetrical cell
configurations is important. In response, we have incorporated key symmetrical cell results into
the main figures (Fig 1e, Fig 2h and 2i, and Figure 3e) in the main text. We believe that this specific
revision item indeed addresses potential readers' concerns more directly regarding the relevance
of the findings to real-world solid-state battery design and operational conditions.

**Modification to the manuscript:**

**Fig. 1. Morphology, microstructure, and fracture statistics of LLZTO solid electrolyte during lithium**
 **dendrite penetration.** (a) Schematic of the cryogenic sample preparation and characterization workflow,
 from the glovebox to cryogenic focused ion beam and scanning/transmission electron microscopy, using
 an inert/vacuum transfer holder. (b) Lithium dendrite growth within LLZTO observed using an in-plane
 cell geometry, which promotes the formation of sharp, straight, through-thickness dendrites. (c)
 Corresponding potential and current profiles during lithium plating. (d) Postmortem cryo-SEM image
 showing a tortuous fracture path induced by dendrite growth. (e) EBSD maps showing lithium plating-
 induced fracture in the LLZTO solid electrolyte under the symmetrical cell geometry (this panel is the only
 dataset in Fig. 1 acquired using the symmetric cell configuration). (f) EBSD map of the fractured region of
 LLZTO under in-plan geometry view, highlighting a mixture of intergranular and transgranular fracture
 modes. (g) Fractions of intergranular and transgranular fractures within LLZTO induced by lithium plating.
 (h) Histogram of intergranular crack deflection angles, showing frequent high-angle deflections. (i) Fracture
 mode map based on phase-field modeling, showing the transition from transgranular to intergranular
 fracture as a function of deflection angle and the ratio of fracture energy between grain interior and grain
 boundary (GB). The fitted experimental data with error bars are taken from Fig. 1h and Fig. S3b.

**Fig. 2. Fractography and elemental distribution at the lithium dendrite tip.** (a) Cryo-SEM image
showing the location of the lithium dendrite tip in the LLZTO solid electrolyte, and the regions selected for
S/TEM lamella preparation, including areas for 3D reconstruction, plan-view, and cross-sectional
observations. (b) Three-dimensional reconstruction of the crack network near the dendrite tip obtained by
cryogenic FIB serial sectioning. (c) Plan-view and (d) cross-sectional observations of the lithium dendrite
within the LLZTO electrolyte, corresponding to the region marked in (a). (e) Cross-sectional observation
of a region located approximately 1 μm ahead of the dendrite tip. (f-g) Corresponding EELS lithium count
maps for the boxed regions in (c-d), confirming lithium presence at the crack tip. ABF-STEM: angular
bright field scanning transmission electron microscopy. (h) Three-dimensional reconstruction of lithium
dendrites in the solid electrolyte grown in the symmetric cell configuration, as illustrated by the schematic.
(i) Selected cross-sectional slices used for the 3D reconstruction shown in (h). Labels #1-#3 mark three
seemingly isolated dendrites in Slice 350, which originally branched from a single major dendrite, as shown
in Slice 1 and Fig. 2h. Note that Fig. 2h and 2i were acquired from the symmetric cell configuration in
contrast to Fig. 2a to 2g.

**Fig. 3. Microstructure of lithium dendrite in LLZTO and phase-field fracturing modeling of lithium**
 **dendrite penetration.** (a) HAADF-STEM image showing a lithium dendrite within the solid electrolyte,
 with nanocracks branching from the main dendrite. (b, c) TKD-SEM maps showing the (b) grain orientation
 and (c) grain reference orientation deviation (GROD) within the lithium dendrite. (d) Point-to-origin
 misorientation profiles along the three lines marked in (c), showing negligible crystal lattice rotation across
 the dendrite except the regions near the lithium/LLZTO interface. (e) **Simulated evolution of the maximum**
 **tensile stress in LLZTO, hydrostatic stress in lithium, and von Mises plastic strain in lithium in three**
 **dimensions. The initial crack length equals to 24 μm .** (f) Stress evolution during lithium dendrite
 propagation. (g) Stress response in lithium and LLZTO as a function of lithium yield strength ranging from
 1 MPa to 125 MPa and a purely elastic case, confirming hydrostatic stress as the dominant fracture-driving
 force. TKD-SEM: Transmission Kikuchi diffraction in the scanning electron microscope. Von Mises plastic
 strain: scalar equivalent measure representing the accumulated plastic deformation from a tensorial plastic
 strain state.

Referee 2 comment 2: The EIS data presented in Figure S5c and S9c (now S10c from second round
revision) for the symmetric Li/LLZTO/Li cells raise significant concerns regarding data quality and
interpretation. The Nyquist plots exhibit an unusual, highly compressed semicircular shape in the
high-to-medium frequency range. The manuscript presents the raw EIS data without
accompanying equivalent circuit fits or any quantitative analysis.

**Authors' response:** We thank the reviewer for this suggestion. We have now included a detailed
analysis for the EIS data by providing the equivalent circuit and corresponding fitting parameters.
As shown in the equivalent-circuit fitting, the interfacial response between lithium and LLZTO
reveals a non-ideal capacitive feature and is described as a constant phase element (CPE) with an
exponent of $a=0.82$. For a constant phase element ($a < 1$), the imaginary component is intrinsically
reduced relative to the real component, leading to a depressed or flattened arc rather than a
perfect semicircle. The CPE behavior of the Li-LLZTO interface is due to a continuous spatially
heterogenous charge and mass transfer.^{8, 9} Such compressed semicircular features have been
widely reported for Li-garnet electrolyte interfaces.¹⁰⁻¹³

**Modification to the manuscript:**

**Figure S5. (a) Experimental setup for electrochemical cycling of the lithium/solid**
**electrolyte/lithium symmetric cell inside a glovebox, with controlled stacking pressure and pre-**
**heating to ensure close contact between the lithium metal and the solid electrolyte. (b) Cell**
**potential response during lithium plating and stripping in a symmetric cell at various current**
**densities, with a fixed areal capacity of 1 mAh/cm² per step. (c) EIS scan before applying bias. (d)**
**Fitting parameters for EIS data. (e) Schematic showing sample preparation of shorted solid**
**electrolyte for SEM and EBSD analysis. (f) Cross-sectional optical microscopy image of a short-**
**circuited solid electrolyte embedded in epoxy. The surface was polished with 320-grit sandpaper.**
**(g) The same sample after further fine polishing to a greater depth using a SiO₂ polishing**
**suspension. Data source: symmetrical cell.**

Before cycle	Q_bulk	a_bulk	R_bulk		Q_gb	a_gb	R_gb		Q_interface	a_interface	R_interface	
Unit	F	/	ohm	ohm*cm ²	F	/	ohm	ohm*cm ²	F	/	ohm	ohm*cm ²
Value	9.73E-12	1	823.8	174.9751	7.00E-09	0.80	302.2	64.2	4.45E-07	0.72	265.0	56.3

After cycle	Q_bulk	a_bulk	R_bulk		Q_gb	a_gb	R_gb		Q_interface	a_interface	R_interface	
Unit	F	/	ohm	ohm*cm ²	F	/	ohm	ohm*cm ²	F	/	ohm	ohm*cm ²
Value	9.43E-12	1	750.0	159.3	1.20E-08	0.75	259.1	55.0	3.00E-07	0.69	246.3	52.3

**Figure S10.** (a) Potential response during lithium plating and stripping in the symmetric cell at
0.6 mA/cm^2 . The cell was shut down after 600 s during the stripping stage. (b) Enlarged view of
the stripping period. (c) Electrochemical impedance spectroscopy results before and after cycling.
(d) Fitting parameters for EIS data. Data source: symmetrical cell.

**Notes for Figure S10.** As shown in the equivalent-circuit fitting, the interfacial response between
lithium and LLZTO shows a non-ideal capacitive feature and is described as a constant phase
element (CPE) with an exponent of $a=0.82$. For a constant phase element ($a < 1$), the imaginary
component is intrinsically reduced relative to the real component, leading to a depressed or
flattened arc rather than a perfect semicircle. The CPE behavior of the Li-LLZTO interface is due
to a continuous spatially heterogenous charge and mass transfer.^{8,9} Such compressed semicircular
features have been widely reported for Li-garnet electrolyte interfaces.¹⁰⁻¹³

Referee 2 comment 3: The manuscript complete lacks of any description or discussion related to
Figure S9 (now Figure S10 from second round revision) in the main text. Figure S9 presents critical
electrochemical data for the symmetric cell. These data are fundamental for establishing the
operational state and stability of the symmetric cell. Specially, the stripping voltage profile
presented in Figure S9b is highly abnormal for a symmetric Li/LLZTO/Li cell and raises serious
concerns regarding the interpretation of the experiment.

**Authors' response:** We thank the reviewer for raising this concern. The behavior observed in Fig.
S10 differs from that in Fig. S5b, which underwent multiple cycling steps and exhibits a symmetric
overpotential. However, the stripping/plating behavior observed in Fig. S10b has been frequently
reported in the cycle prior to short circuiting, such as Fig 6c in Hu *et al*³, Fig 2a in Liu *et al*¹⁴, Fig
S4a in Park *et al*¹², Fig 3b in Yu *et al*¹⁵, Fig 3 in Sharafi *et al*¹⁶, etc.

In Fig. S10, a relatively high current density was intentionally applied, under which lithium
dendrite nucleation and growth are expected to occur. A sharp increase in overpotential (red
curve in Fig. S10a) developed after approximately 2500 s of plating, which is attributed to contact
loss on the stripping side.¹⁷ In the subsequent cycle (blue curve in Fig. S10b), after current reversal,
lithium metal preferentially deposits at pre-existing “hot spots,” leading to accelerated dendrite
growth.^{11, 18} Once dendrites begin to grow within the solid electrolyte, the effective transport
distance for Li⁺ ions is reduced, which explains the observed decrease in potential with increasing
time in Fig. S10b. At the end of the experiment, we intentionally terminated the test before hard
short circuit, unlike in the case shown in Fig. S5b. This precaution was taken because a hard short
circuit can induce localized Joule heating, which may melt lithium dendrites and alter the
microstructure of lithium dendrites as for example reported by Manalastas *et al*.¹⁹

We have added additional discussion items related to Fig. S10 in the revised manuscript to
address the reviewer’s concern.

**Modification to the manuscript:**

Page 9: In the Li/LLZTO/Li symmetric cell configuration, the three-dimensional fractography of
LLZTO was also further analyzed, as presented in Fig. 2h. The electrochemical data that pertain
to the growth of lithium dendrites through the symmetrical cell are shown in Fig. S10. A relatively
high current density was intentionally applied, under which lithium dendrite nucleation and growth
are expected to occur. A sharp increase in overpotential (red curve in Fig. S10a) developed after
approximately 2500 s of plating, which is attributed to contact loss on the stripping side.¹⁷ In the
subsequent cycle (blue curve in Fig. S10b), after current reversal, lithium metal preferentially
deposits at pre-existing “hot spots,” leading to accelerated dendrite growth.^{11, 18} Once dendrites
begin to grow within the solid electrolyte, the effective transport distance for Li⁺ ions is reduced,
which explains the observed decrease in overpotential with increasing time in Fig. S10b. At the
end of the experiment, we intentionally terminated the test before hard short circuit, unlike the case
shown in Fig. S5b. This precaution was taken because a hard short circuit may induce localized
Joule heating, which might melt lithium dendrites and alter the microstructure of the lithium

dendrites.¹⁹ For the lithium dendrite growth through the symmetrical cell configuration, the
reconstructed lithium dendrite morphology shown in Fig. 2h and Fig. S11 exhibits a tortuous
geometry similar to that shown in Fig. 2b. Notably, the seemingly isolated lithium dendrite #1 to
#3 (Fig. 2i, slice 350) originates from a single, relatively continuous dendrite (Fig. 2i, slice 1).
Without three-dimensional reconstruction at cryogenic temperature, such an observation could be
misinterpreted as evidence for isolated lithium nucleation and propagation within the LLZTO
electrolyte.

Referee 2 comment 4: The practicality and novelty of void strategy remain questionable, as raised
by Reviewer 1. The discussion on page 18 acknowledges the trade-off with ionic conductivity but
remains speculative. The suggestion that bilayer electrolyte interfaces could serve as "natural
voids" is interesting but underdeveloped. The void introduction, conductivity loss, and scalability
must be solved for practical application.

**Authors' response:** We appreciate the reviewer's support on this aspect and the continued focus
on the practical implication sides of the void-related dendrite-guidance concept. Similar to
question raised by reviewer 1, we agree that implementing void engineering in the most ideal
thin solid electrolytes may presents certain practical challenges at such small and low-
dimensional scales related to ionic conductivity and scalable fabrication. We think that one
possible way to implement void engineering is to use a bilayer solid state electrolyte system. We
believe that, by controlling particle size, stacking pressure from rolling, sintering temperature and
time, interfaces in bilayer solid electrolytes could be engineered as a purposefully designed
mechanical weak spot to deflect Li dendrite growth. While our proof-of-concept demonstrated in
this work was not evaluated under thin solid electrolytes, we provide the fundamental design
principle to achieve safe operation. Future studies will therefore be required to assess the
scalability of void engineering in thin solid-electrolyte films while maintaining a highly uniform
and ionically conductive state of the material. We are afraid though that this full engineering
implementation assessment is beyond the scope of the current paper.

The revised manuscript explicitly presents the introduction of voids as a proof-of-concept
intended to expand future design options for solid-state batteries. The discussion now primarily
emphasizes the underlying design principle, namely, that introducing local mechanical defects
(such as voids, cracks, or weak interfaces) can influence the lithium dendrite propagation
pathways. For example, interfaces in multilayer solid electrolytes may serve as mechanically weak
regions that promote lithium dendrite deflection while preserving ionic transport and chemical
stability at the lithium/solid-electrolyte interface.

**Modification to the manuscript:**

Page 17: These results reveal that transverse voids could redirect the growth direction of lithium
dendrites by locally modifying the tensile stress distribution within the ceramic electrolyte. This
mechanistic understanding offers a proof-of-concept for implementing defect-based strategies to
control dendrite propagation and suppress short-circuiting in solid-state batteries.

Page 18: Mechanically guided redirection of dendrite propagation: As shown in Fig. 4a-4d,
transverse voids aligned perpendicular to the dendrite propagation direction redirect dendrite
growth paths and thus prevent short-circuiting. This proof of concept shows that introducing local
defects (e.g. voids, cracks, or weak interfaces) can effectively influence the dendrites' propagation
paths. To realize this concept in an thin solid electrolyte separators (ideally down to $\sim 20 \text{ um}^1$),

interfaces in multilayer solid electrolytes could potentially be leveraged as mechanically weak
regions to redirect dendrite propagation.²⁻⁴ This approach provides a fundamental design principle
for mitigating dendrite-induced short-circuiting while preserving overall ionic transport and
chemical stability. Scalable fabrication in such lower-dimensional engineering systems require
further investigated.

Referee 2 comment 5: The authors have conducted extensive new analyses, including residual
stress measurements to demonstrate that the deformation zone surrounding the Vickers indents
has negligible influence on Li dendrite deflection. An indent is a complex mechanical feature
consisting of a plastically deformed core and a surrounding strain-affected zone, which is not
merely a geometric void. The authors rely on phase-field simulations that treat the indent as a
void, but this model does not explicitly incorporate the possible effects of the deformed material
surrounding the indent.

**Authors' response:** We thank the reviewer for the comment regarding the influence of the plastic
zone around the Vickers indent on lithium dendrite deflection. In ductile materials, the region
surrounding an indent typically consists of a plastically deformed core surrounded by a strain-
affected zone and therefore cannot always be treated as a simple geometric void. However,
garnet-type solid electrolytes are brittle polycrystalline ceramics with weak grain boundaries. The
Vickers indent image (Fig. 4c) indicates that the region around the indent is dominated by cracks
emanating from the indented area, rather than by an appreciable plastic zone, which would
otherwise exhibit obvious slip traces. Importantly, EBSD mapping (Fig. S31) around the indent,
both at and below the surface, shows no measurable orientation gradients and no systematic
increase in local misorientation (e.g., kernel average misorientation) in the vicinity of the indent.
These observations suggest that plastic strain accumulation adjacent to the indent is negligible.
Therefore, treating the indent-induced cracks as an effective void in our phase-field framework is
a reasonable approximation for capturing their dominant effect on lithium dendrite penetration
and deflection.

To further address the reviewer's concern that a plastic zone could play a role in lithium dendrite
deflection, we performed additional phase-field modelling by introducing a thin annular shell
surrounding the void (Fig. S35), with a yield stress corresponding to that of the LLZTO solid
electrolyte. The yield stress can be estimated from the indentation hardness using the Tabor
factor $C = \frac{H}{\sigma_y}$. Following Johnson's prediction²⁰ for hardness measured using a conical indenter,
the relationship between hardness and yield stress is given by

$$\frac{H}{\sigma_y} = \frac{2}{3} \ln\left(\frac{E}{\sigma_y} \tan\beta\right),$$

where H is the indentation hardness, E is Young's modulus, and β is the indenter half-angle, which
is 19.7° for a Berkovich indenter. Using the modulus and hardness values obtained from
nanoindentation measurements (Fig. S34), we calculate a Tabor factor $C = 1.43$ which yields a
yield stress of $\sigma_y = H/1.43 = 5.6$ GPa. In the additional phase-field model (Fig. S35), a thin annular
shell with a yield stress of 5.6 GPa was added around the voids. The predicted dendrite deflection
behavior remains unchanged under these conditions.

**Modification to the manuscript:**

Page 16: We note that plastic strain accumulation adjacent to the indent is negligible, as EBSD
mapping around the indent (both at and below the surface as shown in Fig. S31) reveals no
measurable orientation gradients and no systematic increase in local misorientation (e.g., kernel
average misorientation).

Page 22:

**Small scale mechanical testing on LLZTO solid electrolyte**

$\text{Li}_{6.6}\text{La}_3\text{Zr}_{1.6}\text{Ta}_{0.4}\text{O}_{12}$ (LLZTO) solid electrolytes with a thickness of 1 mm were mechanically
ground and polished, with the final step performed using a $0.05\ \mu\text{m}$ alcohol-based colloidal silica
suspension. The samples were immersed in 1 M HCl for 30 s to remove surface contaminants.
Immediately after the acid treatment, the nanoindentation experiments were conducted using an
iMicro nanoindenter (KLA Inc.) under ambient environment. A constant indentation strain rate of
$\dot{\epsilon} = 0.1\ \text{s}^{-1}$ was applied using a Berkovich diamond pyramidal indenter. Four independent sets of
experiments were performed to verify repeatability (Fig. S34), with a total testing duration of
approximately 30 mins. Although a surface carbonate layer can form on LLZTO upon exposure to
ambient conditions, its thickness within ~ 0.5 h of air exposure is expected to be negligible
compared with the indentation depth.¹³

**Figure. S34 Nanoindentation results of polycrystalline LLZTO: (a) load-depth curve; (b) hardness-**
**depth curve; (c) Young's modulus-depth curve; and (d) phase angle-depth curve.**

**Figure S35.** (a, b) Schematic illustration of the introduction of a thin annular shell surrounding the
 void. The shell thickness is 4 μm . The yield stress of the shell layer is 5.6 GPa. (c, d) Phase-field
 simulation of lithium dendrite interaction with (c) circular and (d) transverse voids embedded in
 the solid electrolyte, showing hydrostatic stress in lithium and maximum tensile stress in LLZTO
 during dendrite propagation.

**Notes for Figure. S35.** In the phase-field model, a thin annular shell with a yield stress
 corresponding to that of the LLZTO solid electrolyte. The predicted dendrite deflection behavior
 remains unchanged under these conditions.

The yield stress was estimated from the indentation hardness using the Tabor factor $C = \frac{H}{\sigma_y}$.
 Following Johnson's prediction²⁰ for hardness measured using a conical indenter, the relationship
 between hardness and yield stress is given by

$$\frac{H}{\sigma_y} = \frac{2}{3} \ln\left(\frac{E}{\sigma_y} \tan\beta\right),$$

where H is the indentation hardness, E is Young's modulus, and β is the indenter half-angle, which
 is 19.7° for a Berkovich indenter. Using the modulus and hardness values obtained from
 nanoindentation measurements (Fig. S34), we calculate a Tabor factor $C = 1.43$ which yields a
 yield stress of $\sigma_y = H/1.43 = 5.6$ GPa.

Referee 2 comment 6: My concern, echoed by the other reviewer, persists. While this work
provides a solid characterization of lithium dendrite penetration through cracks (which I
appreciate), it cannot definitively exclude other mechanisms. Real battery scenarios likely vary.
Consequently, the manuscript's conclusion and title ("mechanical origin of...") appear overstated;
this mechanism is likely only one possible origin. Furthermore, the proposed surface-void-
engineering strategy seems impractical, facing significant trade-offs with ionic conductivity and
structural integrity/stability, as acknowledged by the authors in their response. Therefore, I
strongly recommend toning down the conclusions. However, if weakened, I question whether the
manuscript's significance meets Nature's standards relative to other publications in the field.

**Authors' response:** We respectfully and humbly tend to disagree with the reviewer on this
comment. Our experimental results, including cells using plan-view and symmetrical geometry,
clearly demonstrate the hydrostatic pressure inside the lithium dendrite fracture the
polycrystalline garnet solid electrolyte. Moreover, our results display no evidence for isolated
lithium nuclei formation ahead of the dendrite tip in the garnet solid electrolyte, within the
normal battery operating voltage window²² (< 4.5V vs Li/Li+). The mechanism of isolated lithium
formation due to electronic leakage are only observed under high applied bias of 10 V vs Li/Li+ at
the triple junctions of the LLZTO solid electrolyte as reported by Liu *et al*¹⁴ (*Nature Materials* 20.11
(2021): 1485-1490.), while no lithium formation occurred at 2 V or 5 V vs Li/Li+. Thus, our
observations and other references together support our conclusion within the normal battery
operating voltage window.²²

Yet, to comply with the reviewer and avoid any potential overstatement, we have revised the title
and rephrase in the main text (see below), as suggested by the reviewer, to emphasize a
mechanically driven mechanism of lithium dendrite penetration in garnet solid electrolyte rather
than implying universal applicability. Still, the mechanistic understanding (Fig. 1 to Fig. 3) and our
proven strategy to mitigate detrimental lithium dendrite penetration (Fig. 4) by microstructure
engineering and micromechanical considerations (here via indentation induced cracks/voids) are
of high novelty and significance for the battery community.

The reviewer's concern regarding the practicality of void-related strategies has been addressed
separately in our response items above and in the correspondingly revised manuscript.

**Modification to the manuscript:**

**Title :** Mechanically Driven Lithium Dendrite Penetration in Garnet Solid Electrolyte

**Page 2:** Building on the identified mechanically driven mechanism of lithium penetration in garnet
solid electrolyte, we introduce geometrically engineered voids into the electrolyte to redirect
lithium dendrite growth and mitigate short-circuiting.

**Page 4:** Based on the identified **mechanically driven mechanism** of lithium dendrite penetration
**in garnet solid electrolyte**, we propose a mechanics-informed strategy to redirect lithium dendrite
propagation through the introduction of geometrically engineered voids in LLZTO.

**Page 16:** Based upon the identified **mechanically driven mechanism** of lithium dendrite
penetration **in garnet solid electrolyte**, we propose a mitigation strategy that employs cracks to
redirect lithium dendrite growth in a transverse direction (Fig. 4a), aiming to preventing short
circuit.

Reference

- (1) Janek, J.; Zeier, W. G. Challenges in speeding up solid-state battery development. *Nature*
*Energy* **2023**, *8* (3), 230-240. DOI: 10.1038/s41560-023-01208-9.
- (2) Ye, L.; Li, X. A dynamic stability design strategy for lithium metal solid state batteries. *Nature*
**2021**, *593* (7858), 218-222. DOI: 10.1038/s41586-021-03486-3.
- (3) Hu, B.; Zhang, S.; Ning, Z.; Spencer-Jolly, D.; Melvin, D. L. R.; Gao, X.; Perera, J.; Pu, S. D.;
Rees, G. J.; Wang, L.; et al. Deflecting lithium dendritic cracks in multi-layered solid electrolytes.
*Joule* **2024**, *8* (9), 2623-2638. DOI: 10.1016/j.joule.2024.06.024.
- (4) Yu, Z.; Gan, C.; Mijailovic, A. S.; Stone, A.; Hurt, R.; Pernia, C. L.; Xiao, X.; Shi, C.; Sheldon, B.
10 W. Lithium Dendrite Deflection at Mixed Ionic–Electronic Conducting Interlayers in Solid
Electrolytes. *Advanced Energy Materials* **2025**, *15* (13), 2403179. DOI:
<https://doi.org/10.1002/aenm.202403179>.
- (5) Ning, Z.; Li, G.; Melvin, D. L. R.; Chen, Y.; Bu, J.; Spencer-Jolly, D.; Liu, J.; Hu, B.; Gao, X.;
Perera, J.; et al. Dendrite initiation and propagation in lithium metal solid-state batteries. *Nature*
**2023**, *618* (7964), 287-293. DOI: 10.1038/s41586-023-05970-4.
- (6) Swamy, T.; Park, R.; Sheldon, B. W.; Rettenwander, D.; Porz, L.; Berendts, S.; Uecker, R.;
Carter, W. C.; Chiang, Y.-M. Lithium Metal Penetration Induced by Electrodeposition through
Solid Electrolytes: Example in Single-Crystal $\text{Li}_6\text{La}_3\text{ZrTaO}_{12}$ Garnet. *Journal of The Electrochemical*
*Society* **2018**, *165* (16), A3648. DOI: 10.1149/2.1391814jes.
- (7) Xue, D.; Fincher, C.; Fang, R.; Sheldon, B. W.; Chen, L.-Q.; Zhang, S. Dynamic interplay of
dendrite growth and cracking in lithium metal solid-state batteries. *Journal of the Mechanics*
*and Physics of Solids* **2025**, *202*, 106197. DOI: <https://doi.org/10.1016/j.jmps.2025.106197>.
- (8) Lasia, A. The Origin of the Constant Phase Element. *The Journal of Physical Chemistry Letters*
**2022**, *13* (2), 580-589. DOI: 10.1021/acs.jpcllett.1c03782.
- (9) Córdoba-Torres, P.; Mesquita, T. J.; Nogueira, R. P. Relationship between the Origin of
Constant-Phase Element Behavior in Electrochemical Impedance Spectroscopy and Electrode
Surface Structure. *The Journal of Physical Chemistry C* **2015**, *119* (8), 4136-4147. DOI:
10.1021/jp512063f.
- (10) Lee, K.; Kazyak, E.; Wang, M. J.; Dasgupta, N. P.; Sakamoto, J. Analyzing void formation and
rewetting of thin in situ-formed Li anodes on LLZO. *Joule* **2022**, *6* (11), 2547-2565. DOI:
<https://doi.org/10.1016/j.joule.2022.09.009>.
- (11) Krauskopf, T.; Hartmann, H.; Zeier, W. G.; Janek, J. Toward a Fundamental Understanding of
the Lithium Metal Anode in Solid-State Batteries—An Electrochemo-Mechanical Study on the
Garnet-Type Solid Electrolyte $\text{Li}_6.25\text{Al}_0.25\text{La}_3\text{Zr}_2\text{O}_{12}$. *ACS Applied Materials & Interfaces* **2019**,
*11* (15), 14463-14477. DOI: 10.1021/acsami.9b02537.
- (12) Park, R. J. Y.; Eschler, C. M.; Fincher, C. D.; Badel, A. F.; Guan, P.; Pharr, M.; Sheldon, B. W.;
Carter, W. C.; Viswanathan, V.; Chiang, Y.-M. Semi-solid alkali metal electrodes enabling high
critical current densities in solid electrolyte batteries. *Nature Energy* **2021**, *6* (3), 314-322. DOI:
10.1038/s41560-021-00786-w.
- (13) Sharafi, A.; Yu, S.; Naguib, M.; Lee, M.; Ma, C.; Meyer, H. M.; Nanda, J.; Chi, M.; Siegel, D. J.;
Sakamoto, J. Impact of air exposure and surface chemistry on Li– $\text{Li}_7\text{La}_3\text{Zr}_2\text{O}_{12}$ interfacial
resistance. *Journal of Materials Chemistry A* **2017**, *5* (26), 13475-13487, 10.1039/C7TA03162A.
DOI: 10.1039/C7TA03162A.

- (14) Liu, X.; Garcia-Mendez, R.; Lupini, A. R.; Cheng, Y.; Hood, Z. D.; Han, F.; Sharafi, A.; Idrobo, J.
C.; Dudney, N. J.; Wang, C.; et al. Local electronic structure variation resulting in Li 'filament'
formation within solid electrolytes. *Nature Materials* **2021**, *20* (11), 1485-1490. DOI:
<https://doi.org/10.1038/s41563-021-01019-x>.
- (15) Yu, Z.; Gan, C.; Song, S.; Guduru, P.; Kim, K.-S.; Sheldon, B. W. Dendrite suppression in
garnet electrolytes via thermally induced compressive stress. *Joule* **2026**, *10* (1). DOI:
10.1016/j.joule.2025.102232 (accessed 2026/01/22).
- (16) Sharafi, A.; Meyer, H. M.; Nanda, J.; Wolfenstine, J.; Sakamoto, J. Characterizing the Li-
Li₇La₃Zr₂O₁₂ interface stability and kinetics as a function of temperature and current density.
*Journal of Power Sources* **2016**, *302*, 135-139. DOI:
<https://doi.org/10.1016/j.jpowsour.2015.10.053>.
- (17) Lewis, J. A.; Cortes, F. J. Q.; Liu, Y.; Miers, J. C.; Verma, A.; Vishnugopi, B. S.; Tippens, J.;
Prakash, D.; Marchese, T. S.; Han, S. Y.; et al. Linking void and interphase evolution to
electrochemistry in solid-state batteries using operando X-ray tomography. *Nature Materials*
**2021**, *20* (4), 503-510. DOI: 10.1038/s41563-020-00903-2.
- (18) Krauskopf, T.; Richter, F. H.; Zeier, W. G.; Janek, J. Physicochemical Concepts of the Lithium
Metal Anode in Solid-State Batteries. *Chemical Reviews* **2020**, *120* (15), 7745-7794. DOI:
10.1021/acs.chemrev.0c00431.
- (19) Manalastas, W.; Rikarte, J.; Chater, R. J.; Brugge, R.; Aguadero, A.; Buannic, L.; Llordés, A.;
Aguesse, F.; Kilner, J. Mechanical failure of garnet electrolytes during Li electrodeposition
observed by in-operando microscopy. *Journal of Power Sources* **2019**, *412*, 287-293. DOI:
<https://doi.org/10.1016/j.jpowsour.2018.11.041>.
- (20) Johnson, K. L. *Contact mechanics*; Cambridge university press, 1987.
- (21) Fincher, C. D.; Athanasiou, C. E.; Gilgenbach, C.; Wang, M.; Sheldon, B. W.; Carter, W. C.;
Chiang, Y. M. Controlling dendrite propagation in solid-state batteries with engineered stress.
*Joule* **2022**, *6* (12), 2794-2809. DOI: <https://doi.org/10.1016/j.joule.2022.10.011>.
- (22) Puls, S.; Nazmutdinova, E.; Kalyk, F.; Woolley, H. M.; Thomsen, J. F.; Cheng, Z.; Fauchier-
Magnan, A.; Gautam, A.; Gockeln, M.; Ham, S.-Y.; et al. Benchmarking the reproducibility of all-
solid-state battery cell performance. *Nature Energy* **2024**, *9* (10), 1310-1320. DOI:
10.1038/s41560-024-01634-3.
